# Scaling Vision with Sparse Mixture of Experts

**Carlos Riquelme** *
Google Brain

**Joan Puigcerver** *
Google Brain

**Basil Mustafa** *
Google Brain

**Maxim Neumann**
Google Brain

**Rodolphe Jenatton**
Google Brain

**André Susano Pinto**
Google Brain

**Daniel Keysers**
Google Brain

**Neil Houlsby**
Google Brain

## Abstract

Sparsely-gated Mixture of Experts networks (MoEs) have demonstrated excellent scalability in Natural Language Processing. In Computer Vision, however, almost all performant networks are "dense", that is, every input is processed by every parameter. We present a Vision MoE (V-MoE), a sparse version of the Vision Transformer, that is scalable and competitive with the largest dense networks. When applied to image recognition, V-MoE matches the performance of state-of-the-art networks, while requiring as little as *half* of the compute at inference time. Further, we propose an extension to the routing algorithm that can prioritize subsets of each input across the entire batch, leading to adaptive per-image compute. This allows V-MoE to trade-off performance and compute smoothly at test-time. Finally, we demonstrate the potential of V-MoE to scale vision models, and train a 15B parameter model that attains $90.35\%$ on ImageNet.

## 1 Introduction

Deep learning historically shows that increasing network capacity and dataset size generally improves performance. In computer vision, large models pre-trained on large datasets often achieve the state of the art [57, 50, 36, 20, 3]. This approach has had even more success in Natural Language Processing (NLP), where large pre-trained models are ubiquitous, and perform very well on many tasks [48, 18]. Text Transformers [61] are the largest models to date, some with over 100B parameters [9]. However, training and serving such models is expensive [56, 46]. This is partially because these deep networks are typically "dense"– every example is processed using every parameter –thus, scale comes at high computational cost. In contrast, conditional computation [5] aims to increase model capacity while keeping the training and inference cost roughly constant by applying only a subset of parameters to each example. In NLP, sparse Mixture of Experts (MoEs) are gaining popularity [54, 39, 22], enabling training and inference with fewer resources while unlocking trillion parameter models.

In this work, we explore conditional computation for vision at scale. We introduce the Vision MoE (V-MoE), a sparse variant of the recent Vision Transformer (ViT) architecture [20] for image classification. The V-MoE replaces a subset of the dense feedforward layers in ViT with sparse MoE layers, where each image patch is "routed" to a subset of "experts" (MLPs). Due to unique failure modes and non-differentiability, routing in deep sparse models is challenging. We explore various design choices, and present an effective recipe for the pre-training and transfer of V-MoE, notably outperforming their dense counterparts. We further show that V-MoE models are remarkably flexible. The performance vs. inference-cost trade-off of *already trained* models can be smoothly adjusted during inference by modulating the sparsity level with respect to the input and/or the model weights. Also, we open-source our implementation and a number of V-MoE models trained on ImageNet-21k.[2]

---

*These authors contributed equally. Correspondence to { rikel, jpuigcerver, basilm }@google.com

[2] Mixture of experts code and models available at `http://github.com/google-research/vmoe`.

35th Conference on Neural Information Processing Systems (NeurIPS 2021).

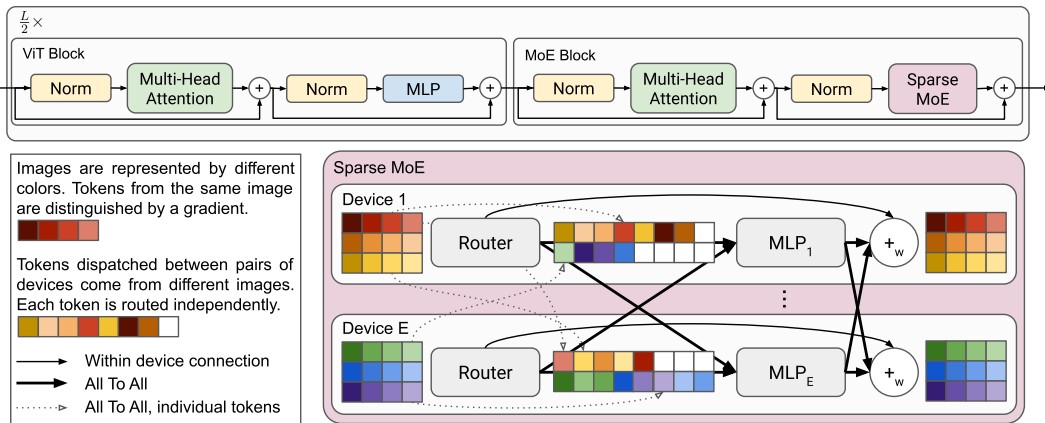

Figure 1: **Overview of the architecture.** V-MoE is composed of $L$ ViT blocks. In some, we replace the MLP with a sparsely activated *mixture* of MLPs. Each MLP (the expert) is stored on a separate device, and processes a fixed number of tokens. The communication of these tokens between devices is shown in this example, which depicts the case when $k = 1$ expert is selected per token. Here each expert uses a capacity ratio $C = \frac{4}{3}$: the sparse MoE layer receives 12 tokens per device, but each expert has capacity for 16 ($\frac{16 \cdot 1}{12} = \frac{4}{3}$; see Section 2.4). Non-expert components of V-MoE such as routers, attention layers and normal MLP blocks are replicated identically across devices.

With V-MoE, we can scale to model sizes of 15B parameters, the largest vision models to date. We match the performance of state-of-the-art dense models, while requiring fewer time to train. Alternatively, V-MoE can match the cost of ViT while achieving better performance. To help control this tradeoff, we propose Batch Prioritized Routing, a routing algorithm that repurposes model sparsity to skip the computation of some patches, reducing compute on uninformative image regions.

We summarize our main contributions as follows:

**Vision models at scale.** We present the Vision Mixture of Experts, a distributed sparsely-activated Transformer model for vision. We train models with up to 24 MoE layers, 32 experts per layer, and almost 15B parameters. We show that these models can be stably trained, seamlessly used for transfer, and successfully fine-tuned with as few as 1 000 datapoints. Moreover, our largest model achieves 90.35% test accuracy on ImageNet when fine-tuned.

**Performance and inference.** We show V-MoEs strongly outperform their dense counterparts on upstream, few-shot and full fine-tuning metrics in absolute terms. Moreover, at inference time, the V-MoE models can be adjusted to either (i) match the largest dense model's performance while using as little as half the compute, or actual runtime, or (ii) significantly outperform it at the same cost.

**Batch Prioritized Routing.** We propose a new priority-based routing algorithm that allows V-MoEs to discard the least useful patches. Thus, we devote less compute to each image. In particular, we show V-MoEs match the performance of the dense models while saving 20% of the training FLOPs.

**Analysis.** We provide some visualization of the routing decisions, revealing patterns and conclusions which helped motivate design decisions and may further improve understanding in the field.

## 2 The Vision Mixture of Experts

We first describe MoEs and sparse MoEs. We then present how we apply this methodology to vision, before explaining our design choices for the routing algorithm and the implementation of V-MoEs.

### 2.1 Conditional Computation with MoEs

Conditional computation aims at activating different subsets of a network for different inputs [5]. A mixture-of-experts model is a specific instantiation whereby different model "experts" are responsible for different regions of the input space [31].

We follow the setting of [54], who present for deep learning a mixture of experts layer with $E$ experts as $\text{MoE}(\mathbf{x}) = \sum_{i=1}^{E} g(\mathbf{x})_i \, e_i(\mathbf{x})$ where $\mathbf{x} \in \mathbb{R}^D$ is the input to the layer, $e_i : \mathbb{R}^D \mapsto \mathbb{R}^D$ is

the function computed by expert $i$, and $g : \mathbb{R}^D \mapsto \mathbb{R}^E$ is the "routing" function which prescribes the input-conditioned weight for the experts. Both $e_i$ and $g$ are parameterized by neural networks. As defined, this is still a dense network. However, if $g$ is sparse, i.e., restricted to assign only $k \ll E$ non-zero weights, then unused experts need not be computed. This unlocks super-linear scaling of the number of model parameters with respect to inference and training compute.

## 2.2 MoEs for Vision

We explore the application of sparsity to vision in the context of the Vision Transformer (ViT) [20]. ViT has been shown to scale well in the transfer learning setting, attaining better accuracies than CNNs with less pre-training compute. ViT processes images as a sequence of patches. An input image is first divided into a grid of equal-sized patches. These are linearly projected to the Transformer's [61] hidden size. After adding positional embeddings, the patch embeddings (tokens) are processed by a Transformer, which consists predominately of alternating self-attention and MLP layers.

The MLPs have two layers and a GeLU [29] non-linearity: $\mathrm{MLP}(\mathbf{x}) = \mathbf{W}_2 \, \sigma_{\mathrm{gelu}}(\mathbf{W}_1 \mathbf{x})$. For Vision MoE, we replace a subset of these with MoE layers, where each expert is an MLP; see Figure 1. The experts have the same architecture $e_i(\mathbf{x}) = \mathrm{MLP}_{\theta_i}(\mathbf{x})$ but with different weights $\theta_i = (\mathbf{W}_1^i, \mathbf{W}_2^i)$. This follows a similar design pattern as the M4 machine translation model [39].

## 2.3 Routing

For each MoE layer in V-MoE, we use the routing function $g(\mathbf{x}) = \mathrm{TOP}_k\left(\mathrm{softmax}\left(\mathbf{W}\mathbf{x} + \epsilon\right)\right)$, where $\mathrm{TOP}_k$ is an operation that sets all elements of the vector to zero except the elements with the largest $k$ values, and $\epsilon$ is sampled independently $\epsilon \sim \mathcal{N}(0, \frac{1}{E^2})$ entry-wise. In practice, we use $k = 1$ or $k = 2$. In the context of the Vision Transformer, $\mathbf{x}$ is a representation of an image token at some layer of the network. Therefore, V-MoE routes patch representations, not entire images.

The difference between previous formulations [54] is that we apply $\mathrm{TOP}_k$ *after* the softmax over experts weights [39], instead of *before*. This allows us to train with $k = 1$ (otherwise gradients with respect to routings are zero almost everywhere) and also performs better for $k > 1$ (see Appendix A).

Finally, we add a small amount of noise with standard deviation $\frac{1}{E}$ to the activations $\mathbf{W}\mathbf{x}$. We empirically found this performed well but that the setup was robust to this parameter. The noise typically altered routing decisions ~15% of the time in earlier layers, and ~2–3% in deeper layers.

## 2.4 Expert's Buffer Capacity

During training, sparse models may favor only a small set of experts [26, 52]. This common failure mode can cause two problems. First, statistical inefficiency: in the limit of collapse to a single expert, the model is no more powerful than a dense model. Second, computational inefficiency: imbalanced assignment of items to experts may lead to a poor hardware utilization.

To combat imbalance and simplify our implementation, we fix the *buffer capacity* of each expert (i.e. the number of tokens that each expert processes), and train our model with auxiliary losses that encourage load balancing. This is essentially the same approach as followed by [54, 39, 22]. In our case, we use slight variants of two of the auxiliary losses proposed in [54], as described in Appendix A.

We define the buffer capacity of an expert ($B_e$) as a function of the number of images in the batch ($N$), the number of tokens per image ($P$), the number of selected experts per token ($k$), the total number of experts ($E$), and the *capacity ratio* ($C$): $B_e = \mathrm{round}\left(\frac{kNPC}{E}\right)$.

If the router assigns more than $B_e$ tokens to a given expert, only $B_e$ of them are processed. The remaining tokens are not entirely 'lost' as their information is preserved by residual connections (the top diagram of Figure 1). Also, if $k > 1$, several experts try to process each token. Tokens are never fully discarded. If an expert is assigned fewer than $B_e$ tokens, the rest of its buffer is zero-padded.

We use the *capacity ratio* to adjust the capacity of the experts. With $C > 1$, a *slack* capacity is added to account for a potential routing imbalance. This is typically useful for fine-tuning when the new data might come from a very different distribution than during upstream training. With $C < 1$, the router is forced to ignore some assignments. In Section 4 we propose a new algorithm that takes advantage of setting $C \ll 1$ to discard the least useful tokens and save compute during inference.

# 3 Transfer Learning

In this section, we first present training different variants of V-MoE on a large dataset (Section 3.2) in order to be used for Transfer Learning afterwards. The ability to easily adapt our massive models to new tasks, using a small amount of data from the new task, is extremely valuable: it allows to amortize the cost of pre-training across multiple tasks. We consider two different approaches to Transfer Learning: linear few-shot learning on fixed representations and full fine-tuning of the model.

## 3.1 Models

We build V-MoE on different variants of ViT [20]: ViT-S(mall), ViT-B(ase), ViT-L(arge) and ViT-H(uge), the hyperparameters of which are described in Appendix B.5. There are three additional major design decisions that affect the cost (and potentially the quality) of our model:

**Number of MoE layers.** Following [39], we place the MoEs on every other layer (we refer to these as V-MoE *Every-2*). In addition, we experimented with using fewer MoE layers, by placing them on the last-$n$ *even* blocks (thus we dub these V-MoE *Last-n*). In Appendix E.1 we observe that, although using fewer MoE layers decreases the number of parameters of the model, it has typically little impact on quality and can speed-up the models significantly, since less communication overhead is incurred.

**Number of selected experts** $k$: The cost of our model does not depend on the total number of experts but the number of *selected* ones per token. Concurrent works in NLP fix $k = 1$ [22] or $k = 2$ [54, 39]. In our case, we use by default $k = 2$ (see Figure 10 in Appendix B for the exploration of different values of $k$), while we found the total number of experts $E = 32$ to be the sweet spot in our setting.

**Buffer capacity** $C$: As mentioned in Section 2.4, we use a fixed buffer capacity. While this is typically regarded as a downside or engineering difficulty to implement these models, we can adjust the *capacity ratio* to control different trade-offs. We can intentionally set it to a low ratio to save compute, using Batch Prioritized Routing (see Section 4). During upstream training, we set $C = 1.05$ by default to give a small amount of slack without increasing the cost noticeably.

Note that for a given trained model, the latter two—$k$ and $C$—can be adjusted without further training, whereas the positioning and quantity of expert layers is effectively fixed to match pre-training.

## 3.2 Data

We pre-train our models on JFT-300M [57], a semi-automatically noisy-labeled dataset. It has ~ 305M training and 50 000 validation images, organised in a hierarchy of 18 291 classes (average 1.89 labels per image). We deduplicate it with respect to all our validation/test sets as in previous efforts [36].[3]

Our few-shot experiments on ImageNet (i.e. ILSVRC2012) use only 1, 5, or 10 shots per class to adapt the upstream model, evaluating the resulting model on the validation set.

We also fine-tuned the pre-trained models on the full training set (ca. 1M images). We report performance in a similar regime for four other datasets in Appendix B.5. Lastly, we explore the ability to fine-tune our large models in the low-data regime by evaluating them on the Visual Task Adaptation Benchmark (VTAB) [69], a diverse suite of 19 tasks with only 1 000 data points per task. As well as natural image classification, VTAB includes specialized tasks (e.g. medical or satellite imagery) and structured tasks (e.g. counting or assessing rotation/distance).

## 3.3 Upstream results

JFT is a multilabel dataset, so we measure model performance via precision@1 (see Appendix B.6 for details). Note that as in previous works [20], hyperparameters were tuned for transfer performance, and JFT precision could be improved at the expense of downstream tasks e.g. by reducing weight decay. Figure 2a shows the quality of different V-MoE and ViT variants with respect to total training compute and time. It shows models that select $k = 2$ experts and place MoEs in the last $n$ even blocks ($n = 5$ for V-MoE-H, $n = 2$ otherwise), but the best results are achieved by V-MoE-H/14 *Every-2* (see Table 2, 14 is the patch size). L/16's are trained for 7 or 14 epochs. See Appendix B.5 for all results.

---

[3]We also checked the effect of deduplication with respect to the ImageNet *training* set, showing negligible (within noise) impact on few-shot results (only 1-shot worsened, see Table 9).

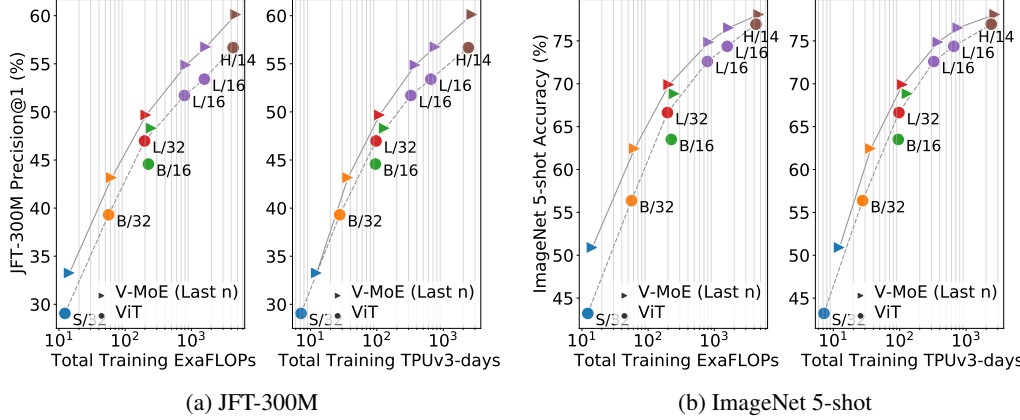

Figure 2: **JFT-300M Precision@1 and ImageNet 5-shot accuracy.** Colors represent different ViT variants, markers represent either standard ●ViT or ▶V-MoEs on the last $n$ even blocks. The lines represent the Pareto frontier of ViT (dashed) and V-MoE (solid) variants.

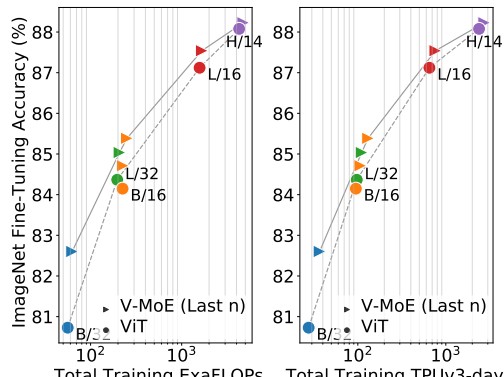

Figure 3: **ImageNet Fine-Tuning Accuracy**. Colors represent different VIT variants, markers represent either standard ●ViT or ▶V-MoEs on the last $n$ even blocks. Lines show the Pareto frontier of VIT (dashed) and V-MoE (solid).

|  | ViT | V-MoE |
|---|---|---|
| L/16 | $76.3_{\pm 0.5}$ | $77.2_{\pm 0.4}$ |
| H/14 | $77.6_{\pm 0.2}$ | $77.8_{\pm 0.4}$ |

Table 1: **VTAB.** Scores and 95% confidence intervals for ViT and V-MoE.

Expert models provide notable gains across all model sizes, for only a mild increase in FLOPs, establishing a new Pareto frontier (gray lines). Alternatively, we can match or improve performance of ViT models at lower cost (e.g. V-MoE-L/16 improves upon ViT-H/14). Similar conclusions hold for training time, which includes communication overhead of dispatching data across devices.

### 3.4 Linear few-shot results

We evaluate the quality of the representations learned using few-shot linear transfer. Given training examples from the new dataset $\{(X, Y)_i\}$, we use the pre-trained model $\mathcal{M}$ to extract a fixed representation $\mathcal{M}(x_i)$ of each image. We fit a linear regression model mapping $\mathcal{M}(x_i)$ to the one-hot encoding of the target labels $Y_i$, following [20] (see [27, Chapter 5] for background).

Figure 2b shows that the upstream gains are preserved under 5-shot ImageNet evaluation, considering both compute and time; in other words, the quality of the representations learned by V-MoE also outperforms ViT models when looking at a new task. Table 2 further shows the results on $\{1, 10\}$-shot for some selected models, and the full detailed results are available in Appendix B.5.

### 3.5 Full fine-tuning results

The typically most performant approach for Transfer Learning [19] consists of replacing the upstream classification head with a new task-specific one and fine-tuning the whole model. Though one may expect that massive models like V-MoEs require special handling for fine-tuning, we broadly follow the standard fine-tuning protocol for Vision Transformers. We use the auxiliary loss during fine-tuning as well, although we observe that it is often not needed in this step, as the router is already well trained. We explore the two sets of tasks considered therein:

Table 2: Main V-MoE & VIT models; Table 8 shows results for additional models and datasets.

| Model | Params | JFT prec@1 | IN/1shot | IN/5shot | IN/10shot | IN/Fine-t. | ExaFLOPs | TPUv3-days |
|---|---|---|---|---|---|---|---|---|
| VIT-H/14 | 656M | 56.68 | 62.34 | 76.95 | 79.02 | 88.08 | 4.27k | 2.38k |
| V-MoE-L/16, Every-2 | 3.4B | 57.65 | 62.41 | 77.10 | 79.01 | 87.41 | 2.17k | 1.20k |
| V-MoE-H/14, Last-5 | 2.7B | 60.12 | 62.95 | 78.08 | 80.10 | 88.23 | 4.75k | 2.73k |
| V-MoE-H/14, Every-2 | 7.2B | 60.62 | 63.38 | 78.21 | 80.33 | 88.36 | 5.79k | 3.47k |
| V-MoE-15B, Every-2 | 14.7B | — | 68.66 | 82.78 | 84.29 | 90.35 | 33.9k | 16.8k |
| NFNet-F4+ [8] | 527M | — | — | — | — | 89.20 | — | 1.86k |
| MPL [49] | 480M | — | — | — | — | 90.20 | — | 22.5k |

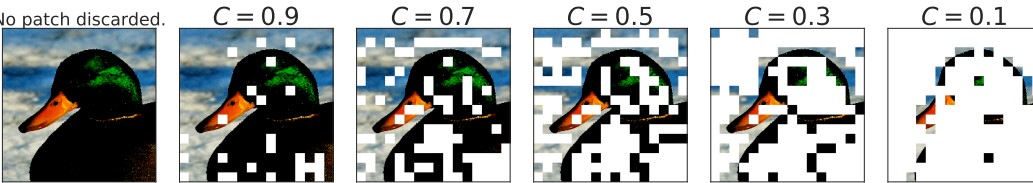

Figure 4: White patches are discarded tokens in the first layer of experts, for different capacities, using Batch Prioritized Routing (Section 4.1) with a V-MoE-H/14. See Appendix D for more examples.

**Full data.** We follow the setup of [20], except that we apply a dropout rate of 0.1 on the expert MLPs (as done in [22]), and we halve the number of fine-tuning steps for all datasets other than ImageNet. Figure 3 shows the results on ImageNet (averaged over three runs). Here, V-MoE also performs better than dense counterparts, though we suspect the fine-tuning protocol could be further improved and tailored to the sparse models. See Table 8 for all details, including results on other datasets.

**Low-data regime.** On the VTAB benchmark, we use a similar setup and hyperparameter budget as [20] (but fine-tune with half the schedule length). Table 1 shows that, while performance is similar for V-MoE-H/14, experts provide significant gains at the ViT-L/16 level, indicating that despite the large size of these models, they can still be fine-tuned with small amounts of data and no further tricks.

### 3.6 Scaling up V-MoE

Finally, we test how well V-MoE can scale vision models to a very large number of parameters, while continuing to improve performance. For this, we increase the size of the model and use a larger pre-training dataset: JFT-3B is a larger version of JFT-300M, it contains almost 3B images and is noisily annotated with 30k classes. Inspired by [68], we apply the changes detailed in Appendix B.3, and train a 48-block V-MoE model, with every-2 expert placement (32 experts and $k = 2$), resulting in a model with 14.7B parameters, which we denote by V-MoE-15B.

We successfully train V-MoE-15B, which is, as far as we are aware, the largest vision model to date. It has an impressive 82.78% accuracy on 5-shot ImageNet and 90.35% when fully fine-tuned, as shown in Appendix B.5, which also includes more details about the model. Training this model required 16.8k TPUv3-core-days. To contextualize this result, the current state of the art on ImageNet is Meta Pseudo-Labelling (MPL) [49]. MPL trains an EfficientNet-based model on unlabelled JFT-300M using ImageNet pseudo-labelling, achieving 90.2% while requiring 22.5k TPUv3-core-days.

## 4 Skipping Tokens with Batch Prioritized Routing

We present a new routing algorithm that allows the model to prioritize important tokens (corresp. patches). By simultaneously reducing the capacity of each expert, we can discard the least useful tokens. Intuitively, not every patch is equally important to classify a given image, e.g., most background patches can be dropped to let the model only focus on the ones with the relevant entities.

### 4.1 From Vanilla Routing to Batch Prioritized Routing

With the notation from Section 2, the routing function $g$ is applied row-wise to a batch of inputs $\mathbf{X} \in \mathbb{R}^{N \cdot P \times D}$. A batch contains $N$ images composed of $P$ tokens each; each row of $\mathbf{X}$ corresponds

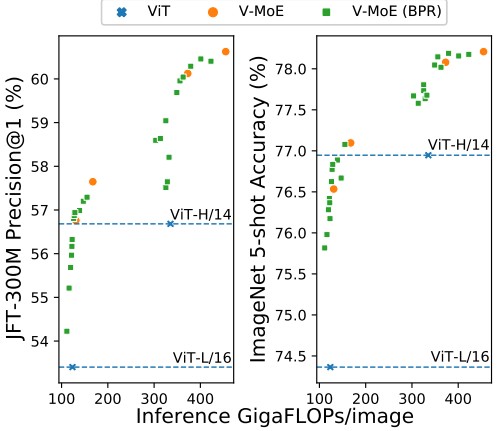

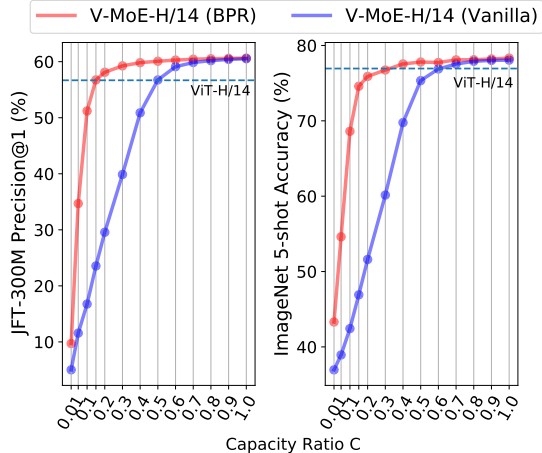

Figure 5: **Reducing compute with priority routing.** Performance vs. *inference* FLOPs for large models. V-MoEs with the original vanilla routing are represented by •, while ■ shows V-MoEs where BPR and a mix of $C \in \{0.6, 0.7, 0.8\}$ and $k \in \{1, 2\}$ are used to reduce compute. ViT models shown as **x**.

Figure 6: **Priority routing works where vanilla fails.** Performance vs. *inference* capacity ratio for a V-MoE-H/14 model with $k = 2$. Even for large $C$'s BPR outperforms vanilla; at low $C$ the difference is stark. BPR is competitive with dense by processing only 15-30% of the tokens.

to the $D$-dimensional representation of a particular token of an image. Accordingly, $g(\mathbf{X})_{t,i} \in \mathbb{R}$ denotes the routing weight for the $t$-th token and the $i$-th expert.

In all routing algorithms considered, for $i < j$, every TOP-$i$ assignment has priority over any TOP-$j$ assignment. The router first tries to dispatch *all* $i^{\text{th}}$ expert choices before assigning *any* $j^{\text{th}}$ choice[4].

Given the TOP-$i$ position, the default—or *vanilla*—routing, as used in [54, 39, 22], assigns tokens to experts as follows. It sequentially goes over the rows of $g(\mathbf{X})$ and assigns each token to its TOP-$i$ expert *when* the expert's buffer is not full. As a result, priority is given to tokens depending on the rank of their corresponding row. While images in a batch are randomly ordered, tokens within an image follow a pre-defined *fixed* order. The algorithm is detailed in Algorithm 1 of Appendix C.

**Batch Prioritized Routing** (BPR). To favour the "most important" tokens, we propose to compute a *priority score* $s(\mathbf{x})$ on each token, and sort $g(\mathbf{X})$ accordingly before proceeding with the allocation. We sort tokens based on their maximum routing weight, formally $s(\mathbf{X})_t = \max_i g(\mathbf{X})_{t,i}$. The sum of TOP-$k$ weights, i.e. $s(\mathbf{X})_t = \sum_i g(\mathbf{X})_{t,i}$, worked equally well. These two simple approaches outperformed other options we explored, e.g., directly parameterising and learning the function $s$.

We reuse the router outputs as a proxy for the priority of allocation. Our experiments show this preserves the performant predictive behaviour of the model, even though the router outputs primarily encode how well tokens and experts can be paired, not the token's "importance" for the final classification task. Figure 4 visualizes token prioritisation with Batch Prioritized Routing for increasingly small capacities. Since all tokens across all images in the batch $\mathbf{X}$ compete with each other, different images may receive different amounts of compute. We summarize BPR in Algorithm 2, in Appendix C.

### 4.2 Skip tokens with low capacity $C$

Batch Prioritized Routing opens the door to reducing the buffer size by smartly selecting which tokens to favor. This can have a dramatic impact in the computational cost of the overall sparse model. We discuss now inference and training results with $C$ defined in Section 2.4 in the regime $C \ll 1$.

**At inference time.** Prioritized routing is agnostic to how the model was originally trained. Figure 6 shows the effect of reducing compute at inference time by using BPR versus vanilla routing, on a V-MoE-H/14 model *trained* using vanilla routing. The difference in performance between both methods is remarkable —especially for $C \leq 0.5$, where the model truly starts fully dropping tokens, as

---

[4]A token may however successfully assign all its TOP-$k$ choices while another may not allocate a single one. This can happen for instance if the latter selects very popular experts that run out of capacity.

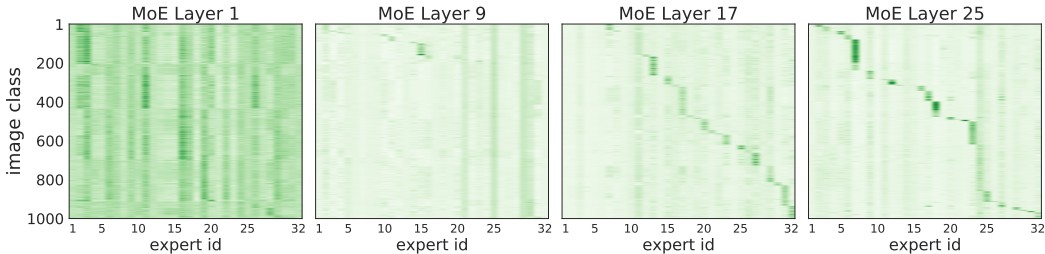

Figure 7: **Deeper routing decisions correlate with image classes**. We show 4 MoE layers of a V-MoE-H/14. The $x$-axis corresponds to the 32 experts in a layer. The $y$-axis are the 1 000 ImageNet classes; orderings for both axes are different across plots. For each pair (expert $e$, class $c$) we show the average routing weight for the tokens corresponding to all images with class $c$ for that particular expert $e$. Figure 29 includes all the remaining layers; see Appendix E.2 for details.

$k = 2$. Also, BPR allows the model to be competitive with the dense one even at quite low capacities. As shown in Figure 5 for V-MoE-L/16 and V-MoE-H/14, Batch Prioritized Routing and low $C$ allow V-MoE to smoothly trade-off performance and FLOPS at inference time, quite a unique model feature. More concretely, Table 10 shows V-MoE models can beat the dense VIT-H performance by using less than half the FLOPs and less than 60% of the runtime. Conversely, we can match the inference FLOPs cost and preserve a one-point accuracy gain in ImageNet/5shot and almost three-point in JFT precision at one (Table 11). Dense models generally require less runtime for the same amount of FLOPs due to the data transfer involved in the V-MoE implementation.

**At training time.** Batch Prioritized Routing can also be leveraged during training. In Appendix C we show how expert models with max-weight routing can match the dense performance while saving around 20% of the total training FLOPs, and strongly outperform vanilla with a similar FLOP budget.

## 5   Model Analysis

Although large-scale sparse MoEs have led to strong performance [22, 39, 54], little is known and understood about how the internals of those complex models work. We argue that such exploratory experiments can inform the design of new algorithms. In this section, we provide the first such analysis at this scale, which guided the development of the algorithms presented in the paper.

**Specialized experts.** Intuitively, routers should learn to distribute images across experts based on their similarity. For instance, if the model had three experts, and the task mainly involved three categories—say animals, cars, and buildings—one would expect an expert to specialize in each of those. We test this intuition, with some obvious caveats: (a) experts are placed at several network depths, (b) $k$ experts are combined, and (c) routing happens at the token rather than the image level.

Figure 7 illustrates how many images of a given ImageNet class use each expert. The plots were produced by running a fine-tuned V-MoE-H *Every-2* model. Interestingly, we saw similar patterns with the upstream model without fine-tuning. Experts specialize in discriminating between small sets of classes (those primarily routed through the expert). In earlier MoE layers we do not observe this. Experts may instead focus on aspects common to all classes (background, basic shapes, colours) - for example, Figure 30 (Appendix E) shows correlations with patch location in earlier layers.

**The value of routers.** After training a sparse MoE, it is natural to study the usefulness of the learned routers, in the light of several pitfalls. For example, the routers may just act as a load balancer if experts end up learning very similar functions, or the routers may simply choose poor assignments. In Appendix E.1, we replace, after training, one router at a time with a uniformly random router. The models are robust to early routing changes while more sensitive to the decisions in the last layers.

**Routing weights distributions.** We analyse the router outputs in Appendix E.3, and observe the distribution of selected weights varies wildly across different mixture of experts layers.

**Changing $k$ at inference time.** We have observed expert models are remarkably flexible. Somewhat surprisingly, sparse models are fairly robust to mismatches between their training and inference configurations. In Appendix E.4, we explore the effect of training with some original value of $k$ while

applying the model at inference time with a different $k' \neq k$. This can be handy to control (decrease or increase) the amount of FLOPs per input in a particular production system.

# 6 Related work

**Conditional Computation.** To grow the number of model parameters without proportionally increasing the computational cost, conditional computation [5, 15, 12] only activates some relevant parts of the model in an *input-dependent* fashion, like in decision trees [7]. In deep learning, the activation of portions of the model can use stochastic neurons [6] or reinforcement learning [4, 17, 53].

**Mixture of Experts.** MoEs [31, 34, 10, 66] combine the outputs of sub-models known as *experts* via a *router* in an input-dependent way. MoEs have successfully used this form of conditional computation in a range of applications [23, 30, 58, 55, 67]. An input can select either all experts [21] or only a sparse mixture thereof as in recent massive language models [54, 39, 22].

**MoEs for Language.** MoEs have recently scaled language models up to trillions of parameters. Our approach is inspired by [54] who proposed a top-$k$ gating in LSTMs, with auxiliary losses ensuring the expert balance [26]. [39] further scaled up this approach for transformers, showing strong gains for neural machine translation. With over one trillion parameters and one expert per input, [22] sped up pre-training compared to a dense baseline [50] while showing gains thanks to transfer and distillation. [40] alternatively enforced a balanced routing by solving a linear assignment problem.

**MoEs for Vision.** For computer vision, previous work on MoEs [21, 2, 25, 1, 63, 47, 64] focused on architectures whose scale is considerably smaller than that of both language models and our model. In DeepMoE [63], the "experts" are the channels of convolutional layers that are adaptively selected by a multi-headed sparse gate. This is similar to [64] where the kernels of convolutional layers are activated on a per-example basis. Other approaches use shallow MoEs, learning a *single router*, either disjointly [25] or jointly [2], together with CNNs playing the role of experts. [1] further have a cost-aware procedure to bias the assignments of inputs across the experts. Unlike shallow MoEs, we operate with up to several tens of routing decisions *per token* along the depth of the model. Scaling up routing depth was marked as a major challenge in [51], which we successfully tackle in our work.

# 7 Conclusions

We have employed sparse conditional computation to train some of the largest vision models to date, showing significant improvements in representation learning and transfer learning. Alongside V-MoE, we have proposed Batch Prioritized Routing, which allows successful repurposing of *model* sparsity to introduce sparsity *with respect to the inputs*. This can be done without further adapting the model, allowing the re-use of trained models with sparse conditional computation.

This has interesting connotations for recent work in NLP using sparse models; recent analysis shows model sparsity is the most promising way to reduce model $CO_2$ emissions [46] and that 90% of the footprint stems from inference costs — we present an algorithm which takes the most efficient models and makes them even *more* efficient without any further model adaptation.

This is just the beginning of conditional computation at scale for vision; extensions include scaling up the expert count, reducing dependency on data and improving transfer of the representations produced by sparse models. Directions relating to heterogeneous expert architectures and conditional variable-length routes should also be fruitful. We expect increasing importance of sparse model scaling, especially in data rich domains such as large scale multimodal or video modeling.

## Acknowledgments and Disclosure of Funding

We thank Alex Kolesnikov, Lucas Beyer and Xiaohua Zhai for providing continuous help and details about scaling ViT models; Alexey Dosovitskiy, who provided some of the pre-trained ViT models; Ilya Tolstikhin, who suggested placing experts only in the last layers; Josip Djolonga for his early review of the manuscript; Dmitry Lepikhin for providing details about the original GShard implementation; Barret Zoph and Liam Fedus for insightful comments and feedback; James Bradbury, Blake Hechtman and the rest of JAX and TPU team who helped us running our models efficiently, and many others from Google Brain for their support.

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
