Table 3: Comparison of routing functions.

| Model | Routing Function | Proposed in | K | prec@1 | ImageNet/1shot | ImageNet/5shot | ImageNet/10shot |
|---|---|---|---|---|---|---|---|
| VIT-S/32 | TOP-K(softmax) | This work | 2 | 34.15 | 38.42 | 53.11 | 56.06 |
| VIT-S/32 | softmax(TOP-K) | [54] | 2 | 33.75 | 35.59 | 50.21 | 53.63 |

Table 4: Simple example ($k = 1$) where average weights are balanced, but Expert 2 is never selected.

| Token | Expert 1 $w_1$ | Expert 2 $w_2$ | Expert 3 $w_3$ | Selected Expert |
|---|---|---|---|---|
| $x_1$ | 0.9 | 0.5 | 0.1 | Expert 1 |
| $x_2$ | 0.1 | 0.5 | 0.9 | Expert 3 |
| $x_3$ | 0.9 | 0.5 | 0.1 | Expert 1 |
| $x_4$ | 0.1 | 0.5 | 0.9 | Expert 3 |
| ... | ... | ... | ... | ... |

# A    Further details about the Vision Mixture of Experts

In this section, we provide additional details about the definition of V-MoE.

The expert function is only computed for the selected top-$k$ experts (while all the routing scores need to be computed first). In particular, we set the expert function to be the Transformer MLP: an MoE layer implements a mixture of the exact same MLP (though each with different weights). For example, when there are 32 experts, we have 32 such MLPs in parallel (each with different weights). The router predicts coefficients which are used to select first and combine then the outputs of multiple such MLPs, e.g. the top-2 for a given example. For $N$ experts, the MoE layer thus consists of $N$ MLPs, and a router which is a single dense layer projecting to size $N$ –one score per expert.

## A.1    Ablation on the modification of the routing function

Our formulation is similar to that in [54], except that we apply the "top $k$" operation *after* normalization of the experts weights, i.e. $\text{TOP}_k$ and softmax are applied in reverse order.

We choose this ordering because the original formulation from [54] cannot be trained easily in the case of $k = 1$; it would lead to zero gradient with respect to $\mathbf{x}$ and $W$ almost everywhere. Moreover, even for $k > 1$, we found our alternative formulation to perform better (see Table 3).

## A.2    Description of the load balancing losses

We describe below the regularizers that we use to enforce a balanced usage of the experts. Those regularizers present slight modifications with respect to their original definitions in [54].

**Importance Loss.** We incentivize a balanced usage of experts via an importance loss. The importance of expert $i$ for a batch of images $\mathbf{X}$ is defined as the normalized routing weight corresponding to expert $i$ summed over images:

$$\text{Imp}_i(\mathbf{X}) \coloneqq \sum_{\mathbf{x} \in \mathbf{X}} \text{softmax}(W\mathbf{x})_i, \tag{1}$$

where $W$ is the layer-specific weight matrix for the router. We use the squared coefficient of variation of the importance distribution over experts, $\text{Imp}(\mathbf{X}) \coloneqq \{\text{Imp}_i(\mathbf{X})\}_{i=1}^{E}$:

$$\mathcal{L}_{\text{Imp}}(\mathbf{X}) = \left( \frac{\text{std}(\text{Imp}(\mathbf{X}))}{\text{mean}(\text{Imp}(\mathbf{X}))} \right)^2 \propto \text{var}(\text{Imp}(\mathbf{X})). \tag{2}$$

[54] proposed a similar loss, while in their case token $\mathbf{x}$ contributed to the importance of expert $i$ in Equation (1) *only* if $i$ was indeed selected for $\mathbf{x}$. We observed some modest empirical benefits thanks to Equation (2).

**Load Loss.** The importance loss seeks to guarantee that all experts have on average similar output routing weights. Unfortunately, it is not difficult to think of routing configurations where these weights are balanced overall, but, still, some small subset of experts get all the assignments (see Table 4).

Ideally, we would like to also explicitly balance the number of assignments. This quantity is discrete; therefore it is not differentiable, and we need to rely on a proxy. Following the proposal in [54], for each expert $i$ and token $\mathbf{x}$, we compute the probability of $i$ being selected —i.e., being among the top-$k$— for $\mathbf{x}$ if we were to re-sample *only* the noise for expert $i$. For simplicity, we slightly modify the definition in [54]. For each token $\mathbf{x}$, we define the score threshold above which experts were selected; this is simply the $k$-th maximum score:

$$\text{threshold}_k(\mathbf{x}) := \max_{k\text{-th}} (W\mathbf{x} + \epsilon), \tag{3}$$

where $\epsilon$ was the noise vector originally sampled during the forward pass. Then, for each expert $i$ we compute the probability of $i$ being above the threshold if we were to only re-sample its noise:

$$p_i(\mathbf{x}) := \mathbf{P}((W\mathbf{x})_i + \epsilon_{\text{new}} \geq \text{threshold}_k(\mathbf{x})) = \mathbf{P}(\epsilon_{\text{new}} \geq \text{threshold}_k(\mathbf{x}) - (W\mathbf{x})_i). \tag{4}$$

The probability is defined over $\epsilon_{\text{new}} \sim \mathcal{N}(0, \sigma^2)$, with $\sigma = 1/E$. The load for expert $i$ over batch $\mathbf{X}$ is:

$$\text{load}_i(\mathbf{X}) = \sum_{\mathbf{x} \in \mathbf{X}} p_i(\mathbf{x}). \tag{5}$$

Finally, the load loss corresponds to the squared coefficient of variation of the load distribution:

$$\mathcal{L}_{\text{load}}(\mathbf{X}) = \left( \frac{\text{std}(\text{load}(\mathbf{X}))}{\text{mean}(\text{load}(\mathbf{X}))} \right)^2, \qquad \text{load}(\mathbf{X}) := \{\text{load}_i(\mathbf{X})\}_{i=1}^E. \tag{6}$$

**Final Auxiliary Loss.** The final auxiliary loss is just the average over both:

$$\mathcal{L}_{\text{aux}}(X) = \frac{1}{2} \mathcal{L}_{\text{imp}}(X) + \frac{1}{2} \mathcal{L}_{\text{load}}(X). \tag{7}$$

The overall loss is: $\mathcal{L}(X) = \mathcal{L}_{\text{classification}}(X) + \lambda \mathcal{L}_{\text{aux}}(X)$, for some hyperparameter $\lambda > 0$. We set $\lambda = 0.01$ in all our experiments, observing that this choice was robust and not sensitive.

Table 5: **Finetuning datasets.**

| Dataset | Num examples | Num classes |
|---|---|---|
| CIFAR10 [37] | 50 000 | 10 |
| CIFAR100 [37] | 50 000 | 100 |
| Oxford Flowers 102 [44] | 1 020 | 102 |
| Oxford-IIT Pet [45] | 3 680 | 37 |
| ImageNet (ILSVRC2012 [16]) | 1 281 167 | 1 000 |

Table 6: **Hyper-parameter values for upstream training on JFT**. Weight decay of 0.1 indicates that this value is applied to all model parameters (including biases), while (0.03, 3) indicates that 0.03 is used for the kernels (the parameters used in all matrix multiplications in dense layers for attention or MLPs, and the initial convolution, but *not* the biases) and 3 for the classification head.

| Variant | JFT-300M Epochs | Optimizer | Base LR | LR decay | Weight Decay |
|---|---|---|---|---|---|
| S/32 | 5 | Adam | $1 \cdot 10^{-3}$ | linear | 0.1 |
| B/16,32 | 7 | Adam | $8 \cdot 10^{-4}$ | linear | 0.1 |
| L/32 | 7 | Adam | $6 \cdot 10^{-4}$ | linear | 0.1 |
| L/16 | {7,14} | Adam | $4 \cdot 10^{-4}$ | linear | 0.1 |
| H/14 | 14 | Adam | $3 \cdot 10^{-4}$ | linear | 0.1 |
| V-MoE-15B | — | Adafactor | $8 \cdot 10^{-4}$ | rsqrt[a] | (0.03, 3) |

---

[a]A linear learning rate cooldown is applied at the end of training.

## B  Transfer Experiment Details

### B.1  Additional fine-tuning datasets

Alongside finetuning on ImageNet (ILSVRC2012[16]), we also train on four other datasets shown in Table 5. For the Visual Task Adaptation Benchmark (VTAB[70]), we finetune on 19 datasets with 1 000 datapoints per class. We refer interested readers to the original work by Zhai et al. [70] for more details, but in brief, the benchmark consists of 3 task categories:

- **Natural tasks** CalTech101 [41] · CIFAR100 [37] · Street View House Numbers (SVHN - [43]) · Describable Textures (DTD - [13]) · Oxford Flowers [44] · Oxford Pets [45] These tasks contain 'classical' natural real-world images obtained with a camera.

- **Specialised tasks** EuroSAT [28] · Diabetic Retinopothy [35] PatchCamelyon [62] · Remote Sensing Image Scene Classification (RESISC - [11]) These are datasets of images which were captured with specialised (medical, satellite etc) photographic equipment.

- **Structured datasets** DeepMind Lab (Object distance prediction - [69]) · SmallNOrb (Azimuth & Elevation prediction - [38] CLEVR (Counting & Distance prediction [33] · Kitti (Vehicle distance prediction [24]) · dSprites (pixel location & orientation prediction - [42]) These assess understanding of scene structure in some way, predominately from synthetic environments. Example tasks include 3D depth estimation and counting.

### B.2  Upstream hyperparameters

We present the architectural details for the upstream models in Table 8 (embedding size—equivalently referred to as hidden size, MLP dimension, number of Transformer blocks, etc.). Table 6 shows the training hyper-parameters for our main models. We use the original setup for each ViT model [20]. However, ViT-S was not formally introduced in [20], and our parameters for ViT-S (dense and sparse) do not match DeiT-Small introduced in [59].

Table 7: **Hyper-parameter values for fine-tuning on different datasets.**

| Dataset | Steps | Base LR | Expert Dropout |
|---|---|---|---|
| ImageNet | 10 000 | {0.0024, 0.003, 0.01, 0.03} | 0.1 |
| CIFAR10 | 2 500 | {0.001, 0.003, 0.01, 0.03} | 0.1 |
| CIFAR100 | 5 000 | {0.001, 0.003, 0.01, 0.03} | 0.1 |
| Oxford-IIIT Pets | 250 | {0.001, 0.003, 0.01, 0.03} | 0.1 |
| Oxford Flowers-102 | 250 | {0.001, 0.003, 0.01, 0.03} | 0.1 |
| VTAB (19 tasks) | 1 250 | 0.001 | 0.1 |

## B.3 Model modifications for scaling to V-MoE-15B

There are many changes to typical dense models which can be applied alongside model sparsity in order to scale models up. In order to scale the base architecture to which we add sparse mixture of expert layers, we make the following changes based on [68]:

- **Low precision**: We use `bfloat16` instead of `float32` to store the gradient moving average.
- **Learning-rate decay:** We replace the linear schedule with an inverse square root schedule (`rsqrt`).
- **Weight decay:** We apply weight decay to the kernel weights in the model with value 0.03 (the parameters used in all matrix multiplications in dense layers for attention or MLPs, and the initial convolution, while biases are not regularized), except for the head kernel where we apply a stronger regularization of 3.0.
- **Model head:** We replace the token head [20]—where the first token is selected—with a new self-attention based head that also includes an additional MLP [68].

The V-MoE-15B model was trained for just under 4 epochs on JFT-3B.

## B.4 Fine-tuning hyperparameters

Table 7 shows the hyperparameters used for finetuning. As discussed, they are broadly identical to those used in the Vision Transformer [20], though with half the schedule length. We also apply expert dropout of 0.1 on the expert MLPs (as suggested in [22]); this did not make a significant difference, typically marginally reducing or improving performance.

We finetuned the V-MoE-15B model on ImageNet at resolution 560x560 for 30 000 steps (i.e., about 6 epochs) with base learning rate 0.006. We used debiased Polyak averaging similar to [20] with momentum 0.999999.

## B.5 Results and details for all models

Table 8: **Upstream, few-shot and downstream performance for dense and sparse models. Architectural details and training costs also provided.** All V-MoE models have $E = 32$ experts and were trained with $C = 1.05$. We specify the number of selected experts per token ($k$), the number of JFT-300M epochs, the number of Transformer blocks ($L$), the number of attention heads ($H$), the patch embedding size ($D$), the hidden size of the MLP, the total number of parameters, the JFT-300M Precision@1 (%), the ImageNet 1, 5 and 10-shot accuracy (%), the fine-tuning accuracy (%) on ImageNet (INet/Ft.), CIFAR10, CIFAR100, Oxford-IIIT Pets, and Oxford Flowers-102; the total training time on a single core of a TPUv3, and the total training compute (in exaFLOPs).

| Name | $k$ | Epochs | Blocks | Heads | Embed. | MLP | Params | JFT-300M | INet/1s | INet/5s | INet/10s | INet/Ft. | CIFAR10 | CIFAR100 | Pets | Flowers | TPUv3-days | ExaFLOPs |
|---|---|---|---|---|---|---|---|---|---|---|---|---|---|---|---|---|---|---|
| ViT-S/32 | — | 5 | 8 | 8 | 512 | 2048 | 36.5M | 29.05 | 29.37 | 43.21 | 46.38 | 73.73 | 97.95 | 87.20 | 91.03 | 96.78 | 7.22 | 12.27 |
| V-MoE-S/32, Last 2 | 1 | 5 | 8 | 8 | 512 | 2048 | 166.7M | 30.93 | 30.65 | 46.06 | 49.47 | 76.32 | 98.05 | 87.93 | 92.62 | 95.88 | 10.83 | 12.50 |
| V-MoE-S/32, Last 2 | 2 | 5 | 8 | 8 | 512 | 2048 | 166.7M | 33.26 | 35.49 | 50.90 | 54.16 | 77.10 | 98.19 | 88.86 | 93.20 | 96.50 | 12.40 | 14.40 |
| V-MoE-S/32, Every 2 | 2 | 5 | 8 | 8 | 512 | 2048 | 296.9M | 34.00 | 37.53 | 51.75 | 54.97 | 77.08 | 98.23 | 88.50 | 94.02 | 97.86 | 17.60 | 16.53 |
| V-MoE-S/32, Last 2 | 5 | 5 | 8 | 8 | 512 | 2048 | 166.7M | 35.49 | 38.77 | 53.60 | 56.94 | 77.59 | 98.25 | 89.25 | 93.26 | 97.31 | 18.49 | 20.44 |
| ViT-B/32 | — | 7 | 12 | 12 | 768 | 3072 | 102.1M | 39.31 | 40.58 | 56.37 | 59.63 | 80.73 | 98.61 | 90.49 | 93.40 | 99.27 | 27.62 | 56.08 |
| V-MoE-B/32, Last 2 | 1 | 7 | 12 | 12 | 768 | 3072 | 395.0M | 41.41 | 44.49 | 60.14 | 63.63 | 81.70 | 98.88 | 91.28 | 94.85 | 99.21 | 30.59 | 56.41 |
| V-MoE-B/32, Last 2 | 2 | 7 | 12 | 12 | 768 | 3072 | 395.0M | 43.17 | 48.04 | 62.45 | 65.72 | 82.60 | 98.67 | 91.47 | 95.25 | 99.21 | 36.80 | 62.75 |
| V-MoE-B/32, Every 2 | 2 | 7 | 12 | 12 | 768 | 3072 | 980.6M | 43.37 | 47.57 | 62.88 | 65.94 | 82.21 | 98.89 | 91.73 | 95.39 | 99.60 | 54.88 | 76.09 |
| V-MoE-B/32, Last 2 | 5 | 7 | 12 | 12 | 768 | 3072 | 395.0M | 43.94 | 49.07 | 63.33 | 66.68 | 82.72 | 98.87 | 91.46 | 95.07 | 99.24 | 49.11 | 81.75 |
| ViT-L/32 | — | 7 | 24 | 16 | 1024 | 4096 | 325.3M | 46.98 | 50.95 | 66.64 | 69.77 | 84.37 | 99.19 | 92.52 | 95.83 | 99.45 | 97.30 | 196.13 |
| V-MoE-L/32, Last 2 | 2 | 7 | 24 | 16 | 1024 | 4096 | 845.8M | 49.68 | 54.52 | 69.90 | 72.80 | 85.04 | 99.24 | 92.50 | 96.34 | 99.08 | 110.65 | 207.94 |
| ViT-B/16 | — | 7 | 12 | 12 | 768 | 3072 | 100.5M | 44.58 | 48.21 | 63.50 | 66.94 | 84.15 | 99.00 | 91.87 | 95.80 | 99.56 | 95.04 | 224.45 |
| V-MoE-B/16, Last 2 | 1 | 7 | 12 | 12 | 768 | 3072 | 393.3M | 47.21 | 51.98 | 67.94 | 70.93 | 84.71 | 99.09 | 92.37 | 96.40 | 99.57 | 106.95 | 225.78 |
| V-MoE-B/16, Last 2 | 2 | 7 | 12 | 12 | 768 | 3072 | 393.3M | 48.31 | 54.92 | 68.84 | 71.81 | 85.39 | 99.21 | 92.78 | 96.56 | 99.63 | 130.86 | 250.70 |
| V-MoE-L/32, Every 2 | 2 | 7 | 24 | 16 | 1024 | 4096 | 3448.2M | 49.31 | 53.61 | 69.21 | 72.02 | 84.81 | 99.18 | 93.02 | 96.32 | 99.33 | 165.51 | 267.10 |
| V-MoE-B/16, Every 2 | 2 | 7 | 12 | 12 | 768 | 3072 | 979.0M | 49.31 | 55.45 | 69.60 | 72.50 | 85.26 | 99.16 | 92.76 | 96.74 | 99.20 | 201.40 | 303.24 |
| ViT-L/16 | — | 14 | 24 | 16 | 1024 | 4096 | 323.1M | 53.40 | 60.25 | 74.36 | 76.62 | 87.12 | 99.33 | 93.93 | 97.12 | 99.63 | 651.26 | 1572.92 |
| V-MoE-L/16, Last 2 | 1 | 14 | 24 | 16 | 1024 | 4096 | 843.6M | 55.80 | 60.53 | 75.81 | 78.00 | 87.47 | 99.39 | 94.39 | 97.09 | 99.39 | 698.14 | 1577.40 |
| V-MoE-L/16, Last 2 | 2 | 14 | 24 | 16 | 1024 | 4096 | 843.6M | 56.76 | 61.46 | 76.53 | 78.64 | 87.54 | 99.29 | 94.19 | 97.37 | 99.58 | 761.27 | 1666.10 |
| V-MoE-L/16, Every 2 | 2 | 14 | 24 | 16 | 1024 | 4096 | 3446.0M | 57.65 | 62.41 | 77.10 | 79.01 | 87.41 | 99.48 | 94.64 | 97.55 | 99.38 | 1205.99 | 2177.14 |
| ViT-H/14 | — | 14 | 32 | 16 | 1280 | 5120 | 655.8M | 56.68 | 62.34 | 76.95 | 79.02 | 88.08 | 99.50 | 94.71 | 97.11 | 99.71 | 2387.99 | 4276.42 |
| V-MoE-H/14, Last 5 | 2 | 14 | 32 | 16 | 1280 | 5120 | 2688.6M | 60.12 | 62.95 | 78.08 | 80.10 | 88.23 | 99.53 | 94.86 | 97.17 | 99.67 | 2735.70 | 4750.73 |
| V-MoE-H/14, Every 2 | 2 | 14 | 32 | 16 | 1280 | 5120 | 7160.8M | 60.62 | 63.38 | 78.21 | 80.33 | 88.36 | 99.58 | 94.91 | 97.45 | 99.68 | 3477.18 | 5795.35 |
| V-MoE-15B | 2 | — | 48 | 16 | 1408 | 6400 | 14705.1M | — | 68.66 | 82.78 | 84.29 | 90.35 | — | — | — | — | 16775.50 | 33943.30 |

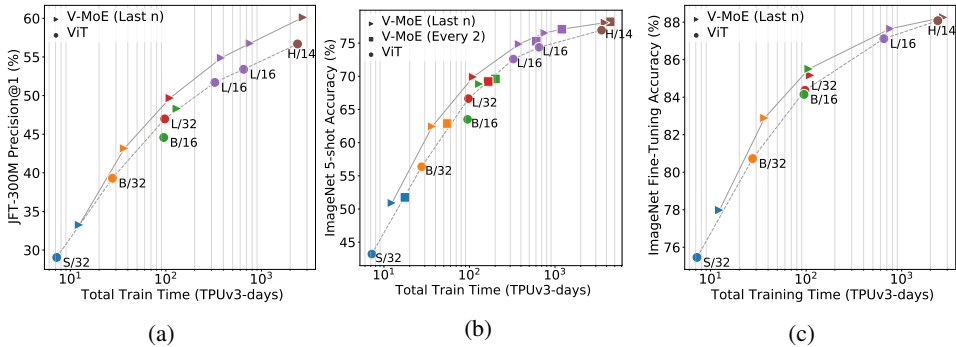

Figure 8: Performance on (a) JFT-300M, (b) ImageNet 5-shot and (c) fine-tuning on full ImageNet achieved by different models as a function of the total training time (TPUv3-core-days). Colors represent different VIT variants, markers represent either standard ●ViT or ▶V-MoEs on the last $n$ even blocks. The lines represent the Pareto frontier of VIT (dashed) and V-MoE (solid) variants.

## B.6  Computing Precision-at-1 on JFT

JFT is multi-label, and it contains a hierarchy of classes. However, for computing precision at one, we ignore this hierarchy: given predictions on an image, we just look at whether the class with highest predicted probability is indeed one of the true labels for the image.

## B.7  Training data deduplication

Table 9 shows the effect of Imagenet deduplication on the training data for fewshot with V-MoE-S/32. Overall, we do not observe a consistent and significant effect after de-duplicating the data. The variance across seeds is notable and—except in the case of IN/1shot—de-duplicated models can outperform (and underperform) the original ones on few-shot evaluation.

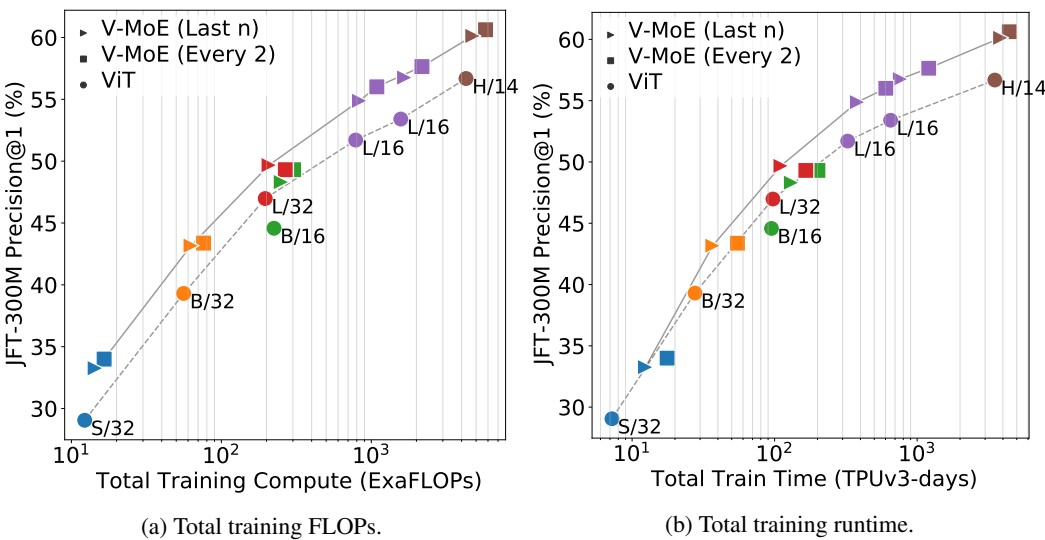

(a) Total training FLOPs.     (b) Total training runtime.

Figure 9: Upstream performance of sparse and dense models. The $x$-axis in (a) shows the total FLOPs required during training, while (b) represents the total training time for identical hardware.

Table 9: **Effect of ImageNet deduplication on the training data for fewshot with V-MoE-S/32.** In order to test the effect of removing some images in the training set that are "close" to some ImageNet ones, we trained three V-MoE-S/32 models —with different seeds— on the de-duplicated dataset, and compare their few-shot performance as shown below. The variance in the results is considerable. The original model dominates on 1-shot, while two out of the three seeds outperform the original model on 5-, 10-, and 25-shot. The de-duplicated dataset contained more images overall, but we limited the training set to the original size (around 305M) and trained for the same epochs.

| Model | Dedup | Seed | IN/1shot | IN/5shot | IN/10shot | IN/25shot |
|---|---|---|---|---|---|---|
| V-MoE-S/32 | No | 0 | **37.53** | 51.75 | 54.97 | 57.44 |
| V-MoE-S/32 | Yes | 0 | 34.07 | 49.34 | 52.21 | 55.11 |
| V-MoE-S/32 | Yes | 1 | 35.63 | 51.95 | 55.79 | 58.19 |
| V-MoE-S/32 | Yes | 2 | 36.72 | **53.09** | **56.50** | **58.84** |

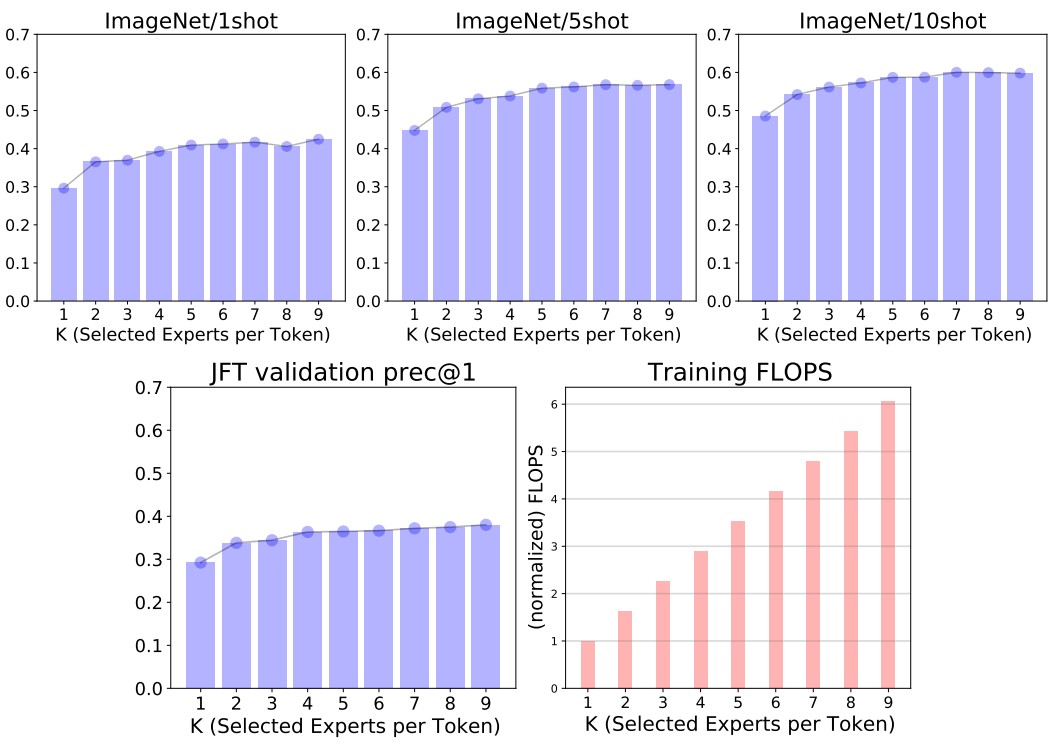

Figure 10: Upstream, few-shot and training FLOPs as a function of $k$ for every-2 V-MoE-S/32.

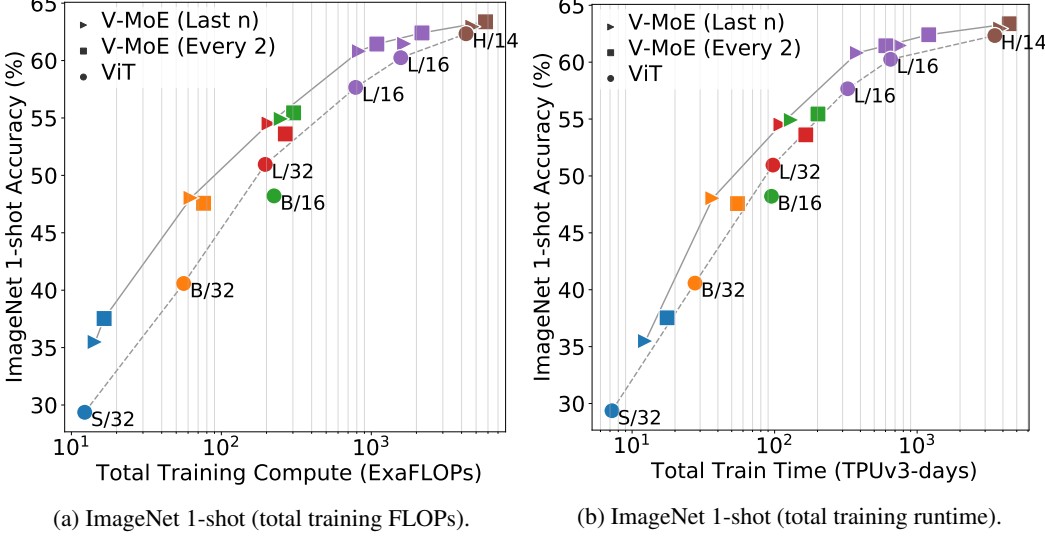

(a) ImageNet 1-shot (total training FLOPs).

(b) ImageNet 1-shot (total training runtime).

Figure 11: ImageNet/1shot performance of sparse and dense models. The $x$-axis in (a) shows the total FLOPs required during training, while (b) represents the total training time for identical hardware.

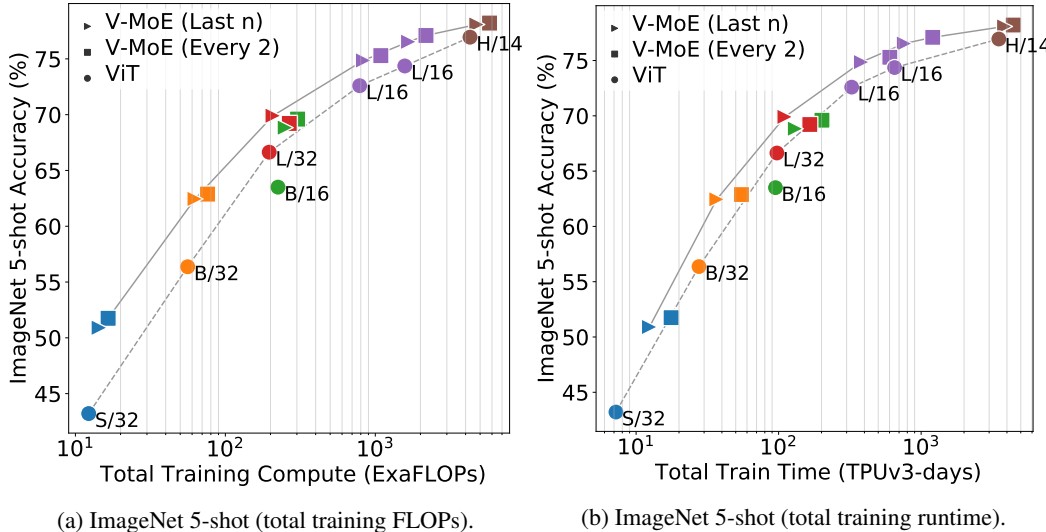

(a) ImageNet 5-shot (total training FLOPs).  (b) ImageNet 5-shot (total training runtime).

Figure 12: ImageNet/5shot performance of sparse and dense models. The $x$-axis in (a) shows the total FLOPs required during training, while (b) represents the total training time for identical hardware.

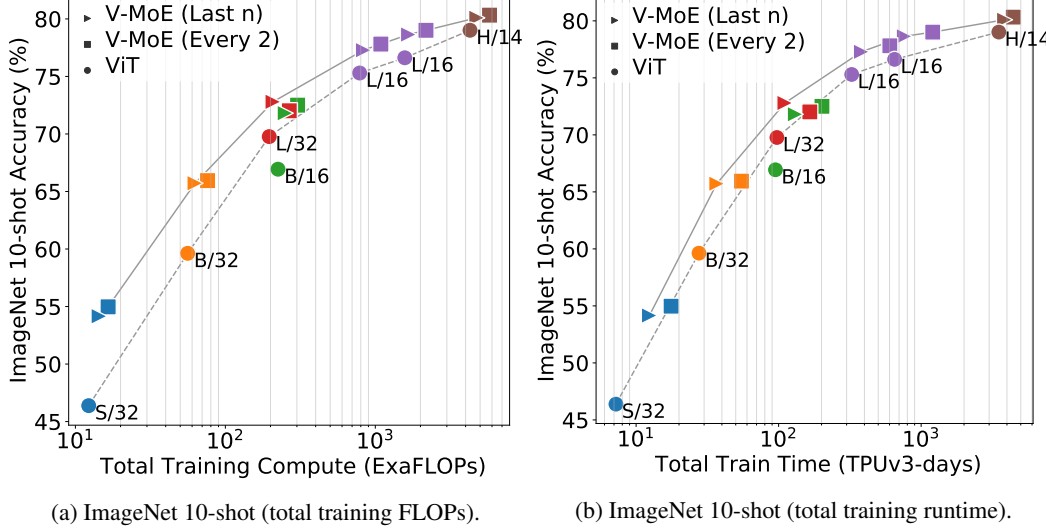

(a) ImageNet 10-shot (total training FLOPs).  (b) ImageNet 10-shot (total training runtime).

Figure 13: ImageNet/10shot performance of sparse and dense models. The $x$-axis in (a) shows the total FLOPs required during training, while (b) represents the total training time for identical hardware.

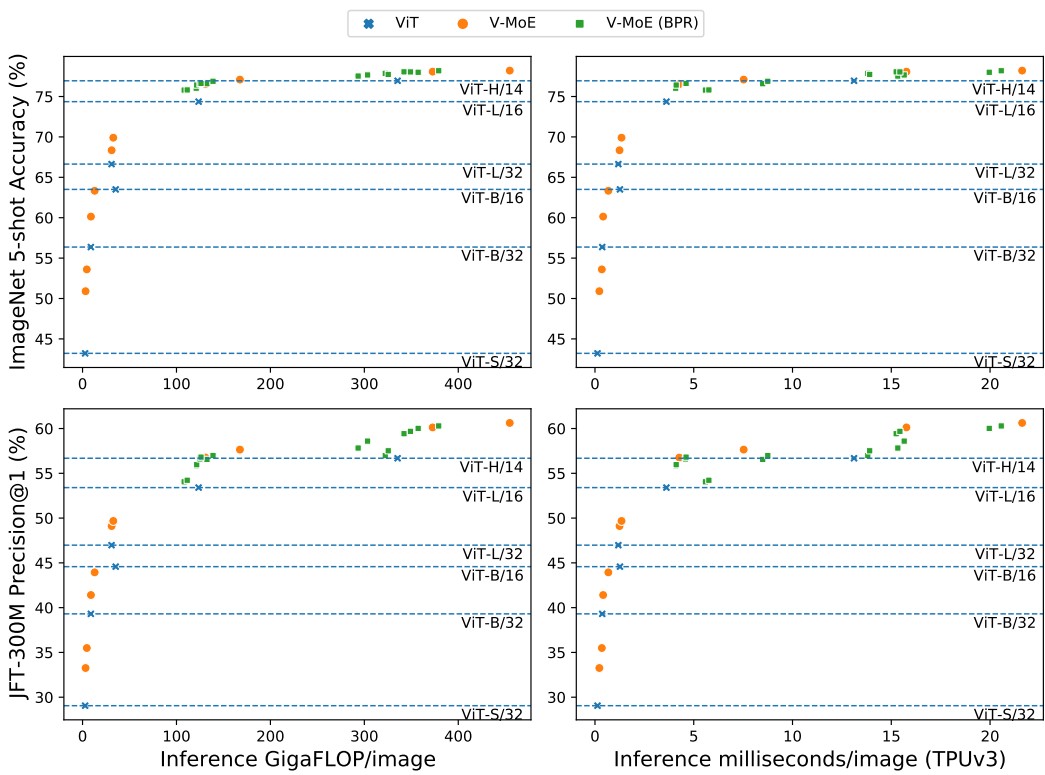

Figure 14: **Reducing compute with priority routing.** Performance vs. *inference* FLOPs and runtime for all models. V-MoEs with the original vanilla routing are represented by ●, while ■ shows V-MoEs where BPR and a mix of $C \in \{0.6, 0.7, 0.8\}$ and $k \in \{1, 2\}$ are used to reduce compute. ViT models shown as x. See Figure 5 for a zoomed-in version on the largest models (versus inference FLOPs).

Table 10: Time and FLOPs *unmatched* inference results for JFT prec@1 and ImageNet 5shot.

| Model | Experts | Routing | JFT prec@1 | INet/5shot | Time[%] | FLOPs[%] |
|-------|---------|---------|------------|------------|---------|----------|
| VIT-H/14 | - | - | 56.68 | 76.95 | 100.00 | 100.00 |
| VIT-L/16 | - | - | 53.40 | 74.36 | 27.58 | 36.83 |
| V-MoE-L/16 | Last-2 | Vanilla | 56.76 | 76.53 | 32.56 | 39.02 |
| V-MoE-L/16 | Every-2 | Vanilla | 57.64 | 77.10 | 57.40 | 49.95 |
| V-MoE-H/14 | Last-5 | Vanilla | 60.12 | 78.08 | 120.22 | 111.12 |
| V-MoE-H/14 | Every-2 | Vanilla | 60.62 | 78.21 | 164.89 | 135.59 |

Table 11: FLOPs *matched* inference results with Batch Prioritized Routing, lower C, and reduced $k$.

| Model | Experts | At Inference | C | JFT prec@1 | INet/5shot | Time[%] | FLOPs[%] |
|-------|---------|--------------|---|------------|------------|---------|----------|
| VIT-H/14 | - | - | - | 56.68 | 76.95 | 100.00 | 100.00 |
| V-MoE-H/14 | Last-5 | k=2 → k=1 | 1.05 | 58.60 | 77.87 | 111.57 | 100.26 |
| V-MoE-H/14 | Last-5 | k=2 → k=1 | 1.25 | 59.21 | 77.59 | 113.67 | 102.53 |
| V-MoE-H/14 | Last-5 | k=2 | 0.5 | 58.61 | 77.92 | 118.14 | 100.02 |
| V-MoE-H/14 | Last-5 | k=2 | 0.6 | 59.42 | 78.05 | 121.68 | 102.30 |
| V-MoE-H/14 | Every-2 | k=2 → k=1 | 1.05 | 59.46 | 77.82 | 134.87 | 100.07 |
| V-MoE-H/14 | Every-2 | k=2 | 0.5 | 59.44 | 77.70 | 155.83 | 100.03 |

# C  Batch Prioritized Routing

## C.1  The Routing Algorithms

---

**Algorithm 1:** Vanilla Routing Allocation

---

**Result:** complete assignment of patches to experts (with some potential dropping)
initialize empty buffers with capacity $B_e$ for all experts $e$ (see Section 2);
**for** $i = 1, \ldots, k$ **do**
    **for** *patch* $p = 1, \ldots, N$ **do**
        $e, w = \text{Router}(\text{TOP} - i \text{ position}, \text{patch } p)$;
        **if** $e$ *is not full* **then**
            add patch $p$ to processing buffer of expert $e$ with weight $w$;
        **else**
            skip $i$-th expert assignment for patch $p$;
        **end**
    **end**
**end**

---

**Algorithm 2:** Batch Prioritized Routing Allocation

---

**Result:** complete assignment of patches to experts (with some potential dropping)
initialize empty buffers with capacity $B_e$ for all experts $e$ (see Section 2);
**for** *patch* $p = 1, \ldots, N$ **do**
    $s(p) = \text{ComputeScore}(\text{patch } p, \text{ Router}(\cdot))$;
**end**
patch ordering $\bar{p} = \text{SortPatches}(\text{scores } s, \text{decreasing} = \text{True})$;
**for** $i = 1, \ldots, k$ **do**
    **for** *patch* $p = (1), \ldots, (N)$ *according to* $\bar{p}$ **do**
        $e, w = \text{Router}(\text{TOP} - i \text{ position}, \text{ patch } p)$;
        **if** $e$ *is not full* **then**
            add patch $p$ to processing buffer of expert $e$ with weight $w$;
        **else**
            skip $i$-th expert assignment for patch $p$;
        **end**
    **end**
**end**

---

We explored a few scoring functions, and concluded that sorting according to the maximum routing weight for each patch $p$ works really well—formally, $s(p) = \max_e w_{e,p}$, where $w_{e,p}$ is the output of the routing function $g$ for patch $p$ and expert $e$ (see Section 4.1). We experimented with the sum of all the TOP-$k$ weights too (rather than just the TOP-1), leading to similar results. Moreover, we tried to directly learn a scoring function. In this case, the router would output $E$ weights per patch (one per expert, jointly normalized by a softmax function) together with the score $s(p)$ —one per patch. We explored a couple of scoring functions (linear + sigmoid, etc), to conclude that the maximum routing weight is quite a good baseline and hard to beat.

A natural extension of this algorithm consists in sorting at the patch-expert assignment level, rather than at the global patch level. The main difference with Algorithm 2 is that the sorting then looks at (patch $p$, TOP$-i$ expert for $p$) scores for $1 \le i \le k$. For example, assume $k = 2$ and we have two patches, $p_1$ and $p_2$. Suppose $p_1$ selects experts $(e_{11}, e_{12})$ with routing weights $(0.7, 0.2)$, while $p_2$ selects $(e_{21}, e_{22})$ with weights $(0.5, 0.4)$. Under Algorithm 2 the order in which patch-expert assignments would be attempted is: $(p_1, e_{11}), (p_2, e_{21}), (p_1, e_{12}), (p_2, e_{22})$. If we use sorting at the patch-expert level, however, we would end up with: $(p_1, e_{11}), (p_2, e_{21}), (p_2, e_{22}), (p_1, e_{12})$. The latter could make more sense as the second assignment for $p_2$ could be more relevant than the second assignment for $p_1$ given their weights. We have not empirically tried this approach, however.

For completeness, we also report another related algorithm we did actually experiment with. We call it *skip-patch*. In this case, we first set a hyper-parameter $S \in (0, 1)$. We will process a fraction $S$ of the patches, and directly **skip** the remaining $1 - S$ fraction. As before, we rank the $N$ patches

according to some scoring function $s(\cdot)$. Then, we directly discard the bottom $(1 - S)\%$ of the patches, and proceed like in Algorithm 2 over the selected $M = SN$ patches. Algorithm 3 formally describes the idea. Going back to our previous example with two patches, if we set $S = 0.5$ there, we will discard $p_2$ altogether, and just process: $(p_1, e_{11}), (p_1, e_{12})$. Note that $S$ and $C$ are two different parameters, and it makes sense to adjust $C$ given $S$ to avoid an excessive FLOPs waste.

---

**Algorithm 3:** Skip-Patch Routing Allocation

---

**Result:** complete assignment of patches to experts (with some **enforced** dropping)
let $S \in (0, 1)$;
initialize empty buffers with capacity $B_e$ for all experts $e$ (see Section 2);
**for** *patch* $p = 1, \ldots, N$ **do**
  $\quad s(p) = \text{ComputeScore}(\text{patch } p, \text{ Router}(\cdot))$;
**end**
patch ordering $\bar{p} = \text{SortPatches}(\text{scores } s, \text{decreasing} = \text{True})$;
patch ordering $\hat{p} = \text{KeepPatches}(\text{TOP} - \text{M}, M = SN, \bar{p})$;
**for** $i = 1, \ldots, k$ **do**
  $\quad$ **for** *patch* $p = (1), \ldots, (M)$ *according to* $\hat{p}$ **do**
    $\quad\quad e, w = \text{Router}(\text{TOP} - i \text{ position, patch } p)$;
    $\quad\quad$ **if** $e$ *is not full* **then**
      $\quad\quad\quad$ add patch $p$ to processing buffer of expert $e$ with weight $w$;
    $\quad\quad$ **else**
      $\quad\quad\quad$ skip $i$-th expert assignment for patch $p$;
    $\quad\quad$ **end**
  $\quad$ **end**
**end**

---

## C.2 Applied during Inference

An appealing property of the algorithms introduced in the previous section is that they are agnostic to how the model was originally trained. Indeed, we first show the effect of reducing compute at inference time by using Batch Prioritized Routing, Algorithm 2, on models trained using Algorithm 1. Note the model parameters are identical in both cases, including the router parameters –we are only applying the model at inference, no further learning is involved–, but we apply different routing strategies. Overall, we observe that discarding patches at random (as Algorithm 1 effectively does) leads to a steep loss of performance when we only keep a small percentage of the patches, as one could expect. On the other hand, if we process the "right" patches —via Algorithm 2— the performance is surprisingly robust as long as we keep up to around 20% of the patches.

Figure 15 shows the inference performance as a function of $C$ for the main every-2 expert models with $k = 2$, under Algorithm 2. We observe performance decreases slowly and smoothly as we constrain more and more the amount of patches experts can process.

Next we compare the inference performance of Algorithms 1 and 2. Results for V-MoE-H/14 are presented in Figure 16, V-MoE-L/16 in Figure 17, V-MoE-B/16 in Figure 18, and V-MoE-S/32 in Figure 19. In all cases we see the same clear trend. By definition of Algorithms 1 and 2, when $k = 2$, if $C \geq 0.5$, then every patch has a decent change of getting its TOP-1 expert processed if routing is balanced. Therefore, the most interesting regime here is $C < 0.5$. In that case, we see an enormous gap in performance between Algorithms 1 and 2, showing that choosing the right patches really pays off. Moreover, in most cases, using 15% of the patches ($C = 0.15$) is enough to match the upstream performance of the dense model. For the few-shot representations, between 20% and 30% of the patches is usually enough.

Overall, we consider the flexibility provided by Algorithm 2 to be quite a remarkable property of expert models. Once trained, they allow for a smooth trade-off between performance and compute, with no further training or adjustment needed. This can be certainly useful in a practical setting where the use-case may determine the available resources and constraints at hand.

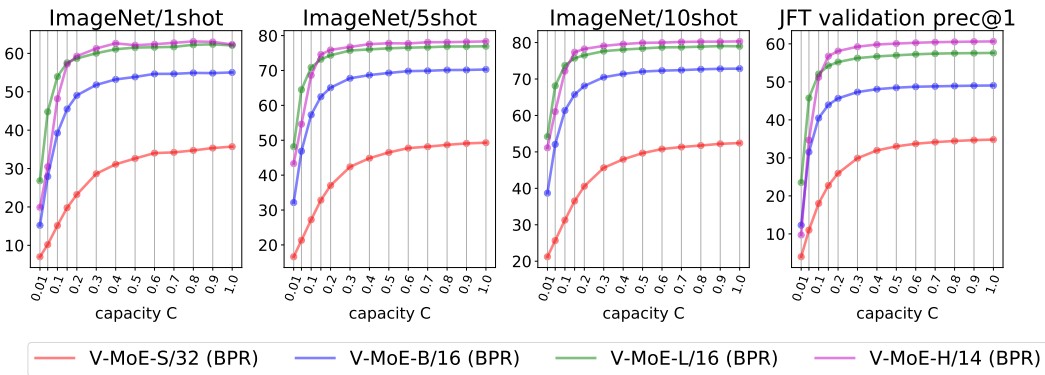

Figure 15: Inference performance for various every-2 V-MoE models with $k = 2$ for different capacities. We show Batch Prioritized Routing.

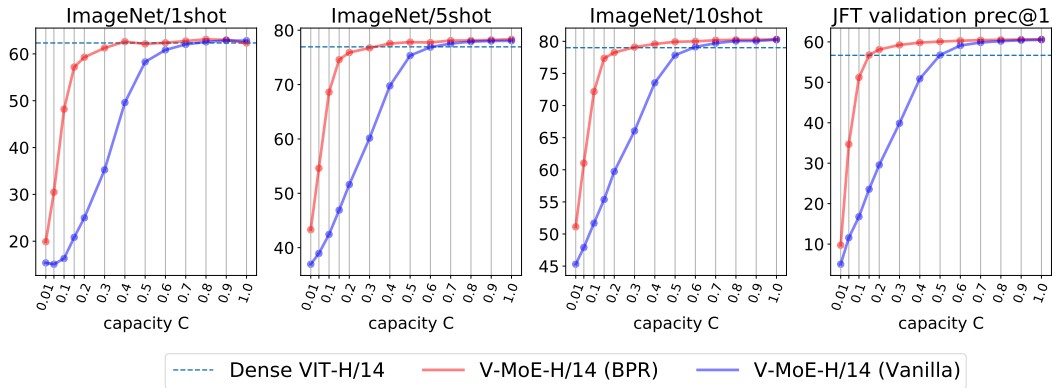

Figure 16: Inference performance for every-2 V-MoE-H/14 model with $k = 2$ for different capacities. We show Batch Prioritized Routing versus vanilla routing.

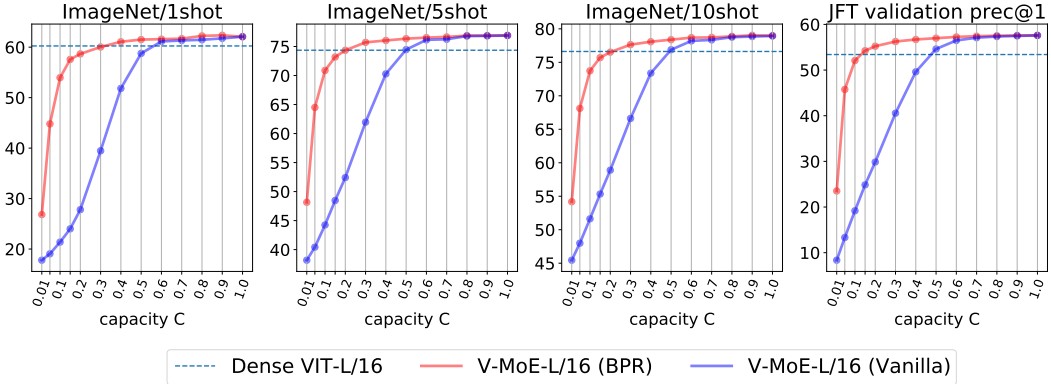

Figure 17: Inference performance for every-2 V-MoE-L/16 model with $k = 2$ for different capacities. We show Batch Prioritized Routing versus vanilla routing.

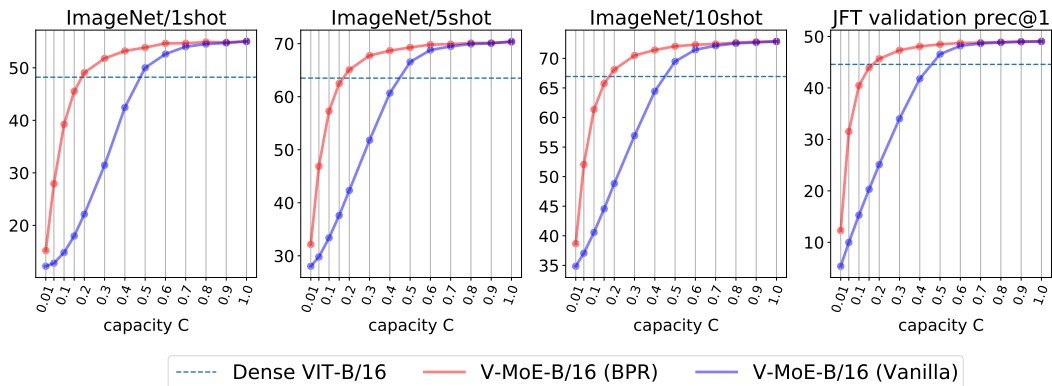

Figure 18: Inference performance for every-2 V-MoE-B/16 model with $k = 2$ for different capacities. We show Batch Prioritized Routing versus vanilla routing.

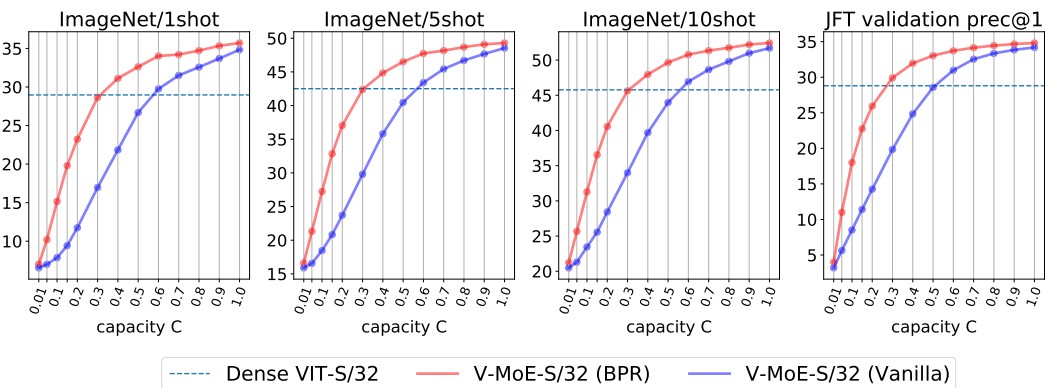

Figure 19: Inference performance for every-2 V-MoE-S/32 model with $k = 2$ for different capacities. We show Batch Prioritized Routing versus vanilla routing.

## C.3 Applied during Training

The previous subsection explored applying priority routing during inference to a pre-trained model. A natural extension consist in directly training a model with Algorithm 2 from scratch. By forcing experts to work with a small buffer or capacity ratio (i.e. $C \ll 1$), we can save substantial training FLOPs while hopefully still get decent performance improvements with respect to dense models.

We show results for three models: V-MoE-S/32, V-MoE-B/32, and V-MoE-L/32. For completeness, we compare Algorithms 1 and 2. In all cases we see strong improvements when training with Algorithm 2. When we use full capacity ($C \geq 1.0$), however, we expect both algorithms to behave in a fairly similar fashion, as no dropping is needed as long as routing is reasonably balanced.

Figures 20 and 21 show V-MoE-S/32 with $k = 1$ and $k = 2$ respectively. We are able to match the dense upstream performance with around 80% of the training FLOPs in both cases. Also, around 85 and 80% of the training FLOPs are enough to match the few-shot evaluation performance in each case. Overall, we can save 20% of the FLOPs while training a small model like V-MoE-S/32.

Figures 22 and 23 show V-MoE-B/32 with $k = 1$ and $k = 2$ respectively. Again, with at most 80% of the training FLOPs the expert models match the upstream performance of its dense counterpart. Also, we can save around 10% of the training FLOPs while keeping or improving the few-shot representation quality.

Finally, Figures 24 and 25 presents the results for VIT-L/32 with $k = 1$ and $k = 2$. Remarkably, between 70 and 75% of the training FLOPs are enough to mimic the upstream dense performance. Note that, when $k = 2$, the lowest capacity ($C = 0.1$) already outperforms the dense upstream precision. The expert model is also able to deliver identical few-shot performance while saving more than 20% of the training FLOPs.

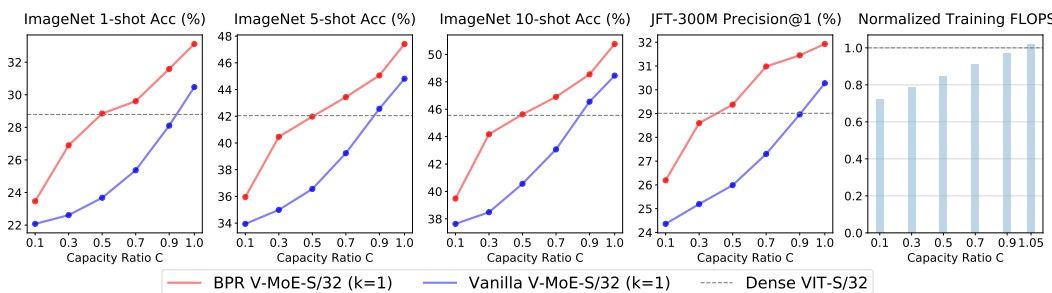

Figure 20: Training with Batch Prioritized Routing. Model: V-MoE-S/32, $k = 1$. Mean over 4 seeds.

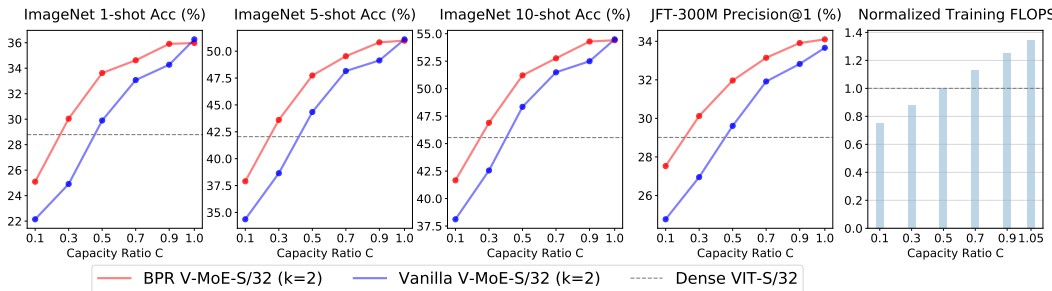

Figure 21: Training with Batch Prioritized Routing. Model: V-MoE-S/32, $k = 2$. Mean over 4 seeds.

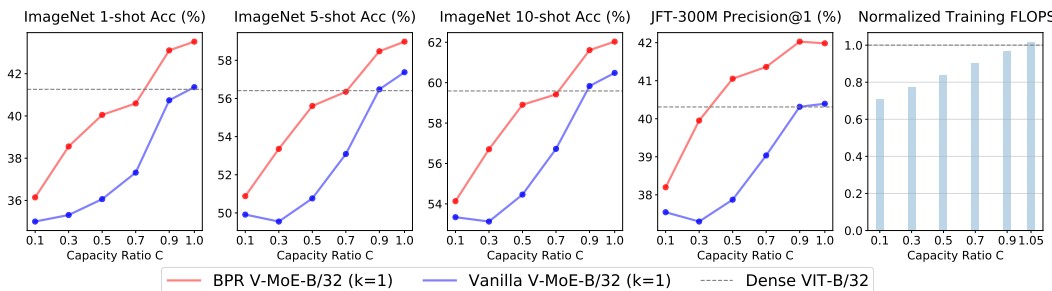

Figure 22: Training with Batch Prioritized Routing. Model: V-MoE-B/32, $k = 1$. Mean over 4 seeds.

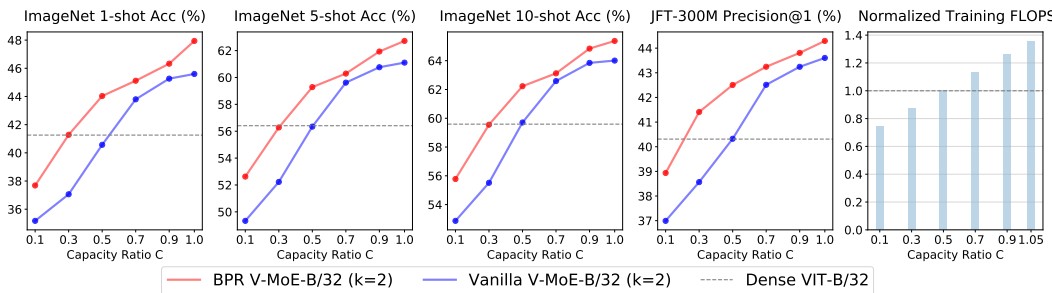

Figure 23: Training with Batch Prioritized Routing. Model: V-MoE-B/32, $k = 2$. Mean over 4 seeds.

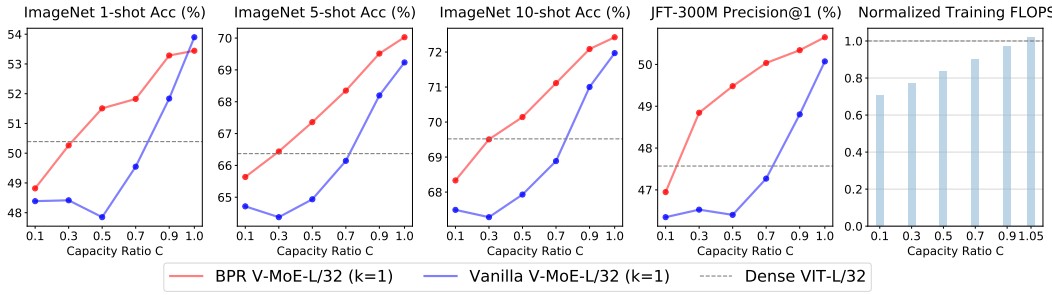

Figure 24: Training with Batch Prioritized Routing. Model: V-MoE-L/32, $k = 1$. Mean over 4 seeds.

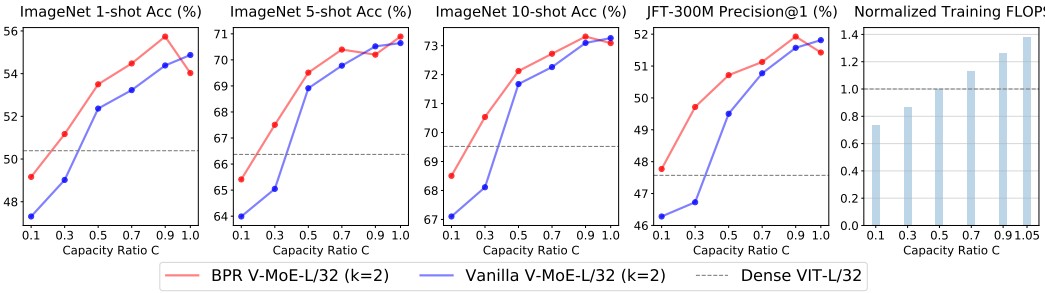

Figure 25: Training with Batch Prioritized Routing. Model: V-MoE-L/32, $k = 2$. Mean over 4 seeds.

## C.4  Applied during Fine-tuning

We also investigate the effect of using the max-routing algorithm in fine-tuning. We consider V-MoE-S/32 models pre-trained at various capacities both with and without Batch Prioritized Routing. We fine tune them on ImageNet to see the effect of priority routing during downstream fine-tuning and inference. This is shown in Figure 26.

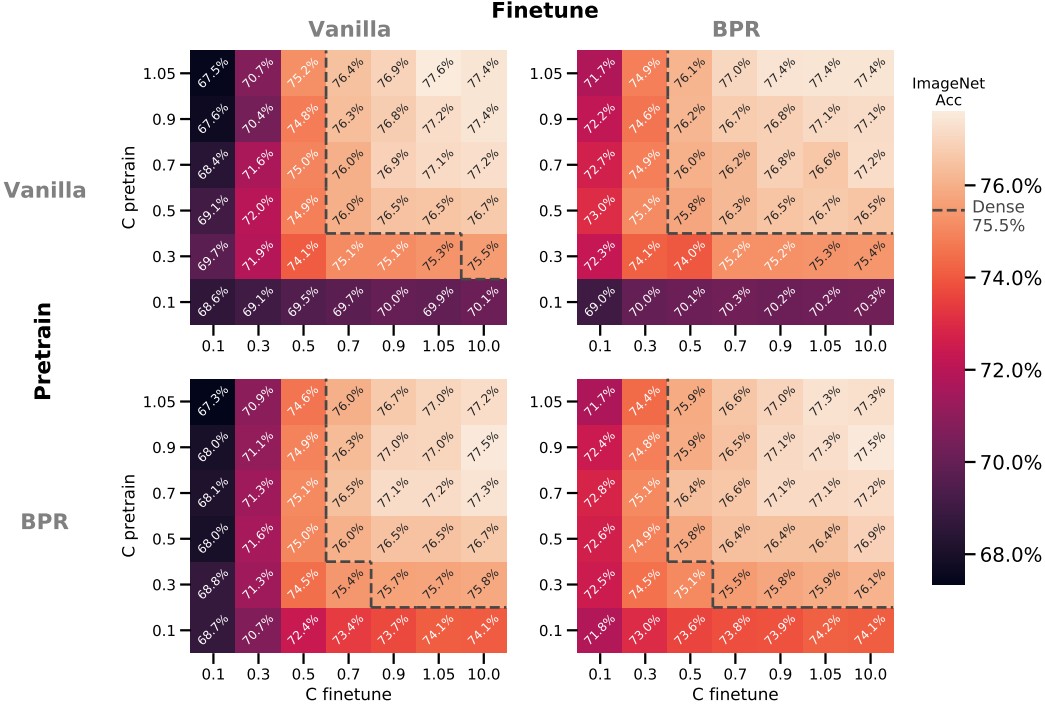

Figure 26: Fine-tuning with Batch Prioritized Routing. Model: V-MoE-S/32, $k = 2$.

There are a few conclusions that can be garnered:

- Downstream fine-tuning results are significantly impacted by capacity, with accuracy reducing from 77.4% to 68.6% by reducing capacity to 0.1.

- Batch Prioritized Routing can recover some of this performance drop; if it is applied during pre-training and fine-tuning, accuracy increases to 71.8% at the same capacity.

- It is more important to retain high capacity during fine-tuning than while pre-training. For example, with priority routing applied both at downstream and upstream, $C = 1.05$ during pre-training with $C = 0.1$ during fine-tuning has accuracy 71.7%, but the inverse is significantly better with accuracy 74.1%. In both cases, priority routing is key to ameliorating the effect of low capacity during fine-tuning and pre-training.

# D    Examples of patch dropping

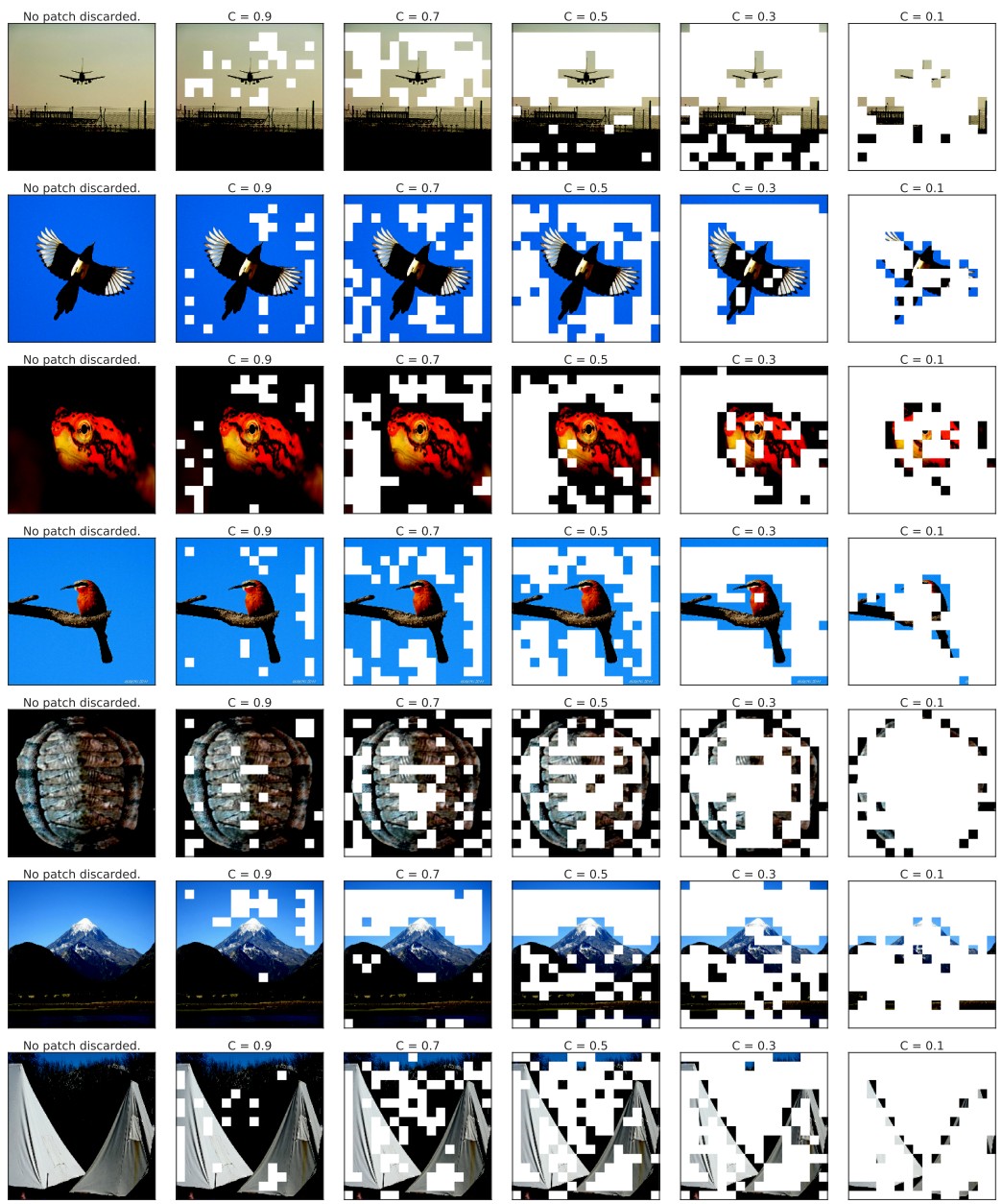

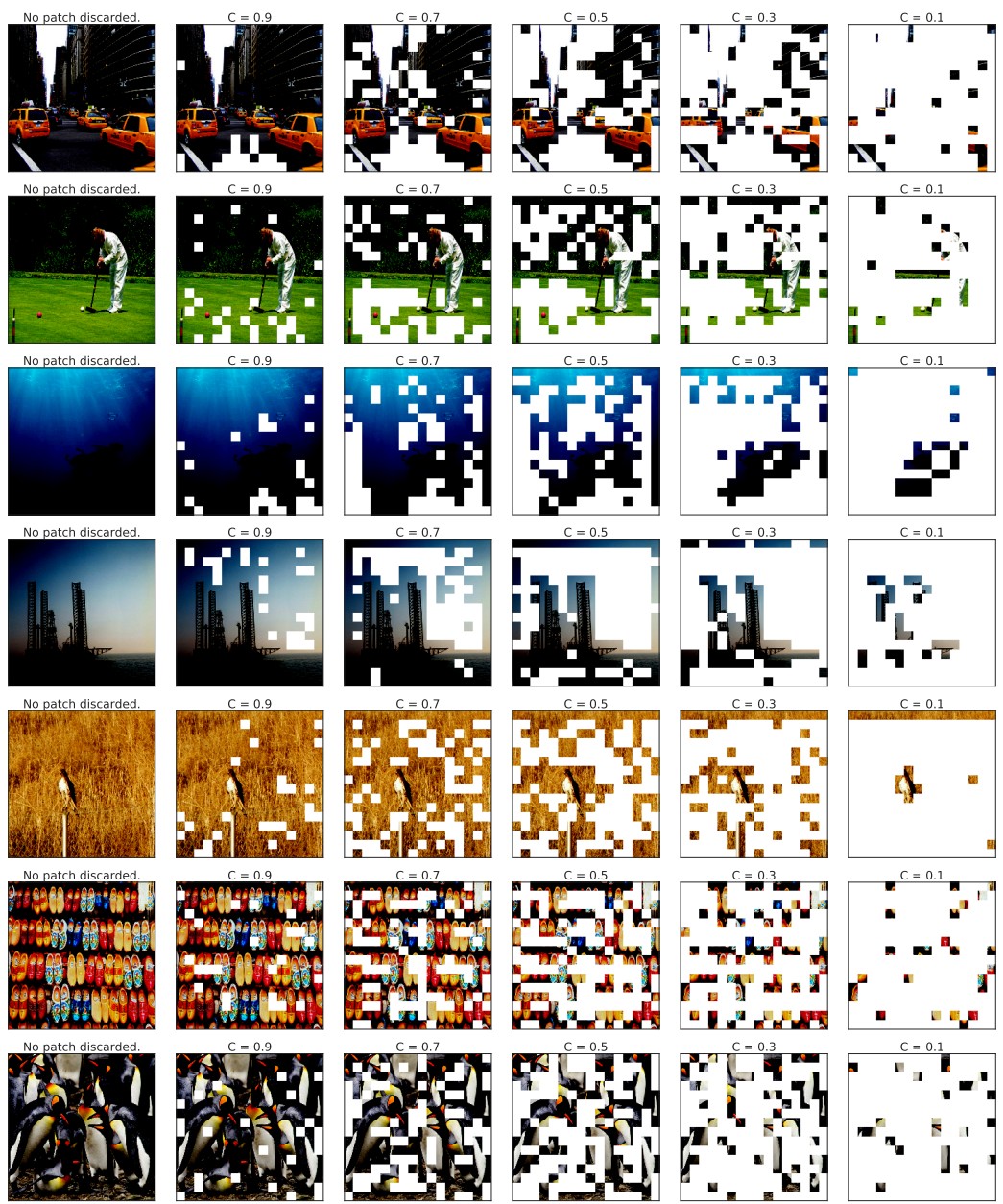

# E    Model Analysis

Several previous works have proposed deep models based on mixture of experts; most of them have also presented promising results. Unfortunately, despite the current excitement regarding this set of techniques, little is indeed known about how these complex models internally work. Exploratory experiments that shed light into the mechanics of routers and expert specialization could inform new algorithms. We try to provide the first such analysis here, which we actually found useful to develop some of the algorithms presented in the paper.

## E.1    The value of routers

The most natural question to ask after training a sparse model is whether the learned routers are doing something useful. There are several potential ways things could go wrong. For example, the router could just become a load balancer if experts end up implementing very similar functions. Alternatively, the router may simply choose sub-optimal assignments. As a first test, we replace one router at a time with a uniformly random router. For this, we take a pre-trained model –in particular, a V-MoE-L/16 with $k = 2$–, and re-evaluate its upstream and few-shot performance when perturbing the routers. Figure 27 contains the results. In red, we show the original performance for the pre-trained model —that is, when applying all the learned routers. We also show the impact of replacing each router independently and in isolation with a uniformly random router –the layer ID is shown in the $x$-axis. In particular, the new router samples the weights in a white Gaussian fashion, so every pair of experts is equally likely to be the TOP-$k$ choice for any given input. We also tried to randomly permute the output weights —so to avoid a distributional shift in applied routing weights—and it worsened results.

Overall, we observe that the last two layers –21 and 23– provide an essential routing for the upstream model to work well (validation precision at 1 in JFT). We have seen a similar pattern in other models. Interestingly, the previous to last MoE layer (21-th in this case) is the one where getting the routing right is the most important. The model is robust to mis-routing at most intermediate layers—layer 9 is an exception here. This observation motivated us into trying to train sparse models with MoE layers only at the very end—21 and 23, for example—with excellent results (and computational savings).

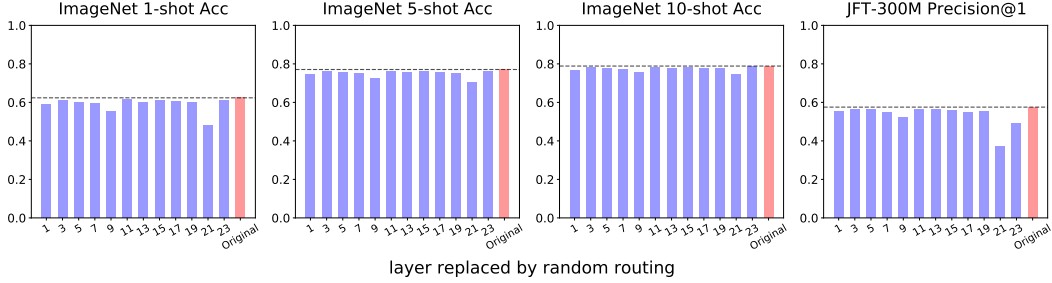

Figure 27: Replace one layer at a time by a random router for V-MoE-L/16.

After analyzing the results in Figure 27, a natural follow up question is whether the model is robust to compounded mis-routing? We answer this question by looking at what happens when we replace a number of consecutive MoE layers with uniformly random routers. Figure 28 shows the outcome. We start from the bottom MoE layer, and for every MoE layer $i$ in the network, we evaluate the model where routers in 1 to $i$ layers (both included) act randomly. Unfortunately, in this case, performance drops quickly as one would expect. Tokens are following random walks (if we ignore capacity issues) up to some point, and then using the correct remaining routers. If the random walk is long enough, the performance is severely degraded. We conclude the token paths in a trained model are far from random or meaningless.

## E.2    Specialized experts

In Figure 29 we show results for a massive model with 24 MoE layers, each of them with 32 experts. After training the model on JFT and fine-tuning it on ImageNet, we did forward passes (up to the pre-logits) with ImageNet images. Each plot corresponds to one MoE layer, in increasing order. The

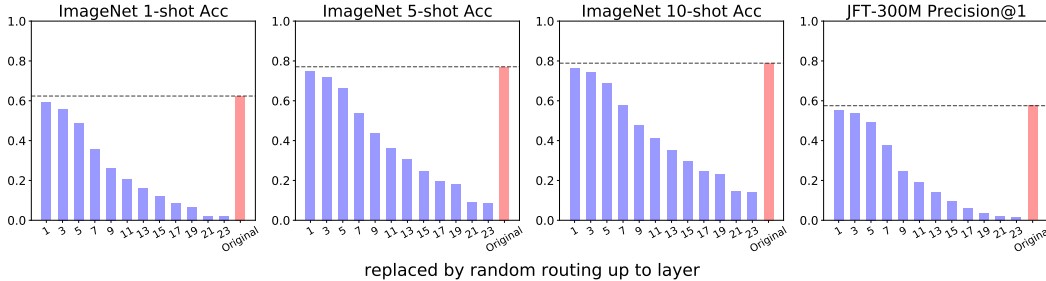

Figure 28: Replace all layers up to a given one by random routers for V-MoE-L/16.

$x$-axis corresponds to the 32 experts per layer, and the $y$-axis are the 1000 ImageNet classes (in different adjusted orders per plot; i.e., class 5 in layers $i$ and $j$ are generally different for $i \neq j$). For each pair (expert $e$, class $i$) we show the average routing weight for the patches corresponding to all images with class $i$ for that particular expert $e$. Intuitively, this is a proxy for how much images of class $i$ activate and use expert $e$. Figure 29 shows strong expert-class correlations for the last few layers. In other words, it seems experts specialize in discriminating between a small set of classes (those primarily routed through the expert). In the initial MoE layers, however, we do not observe such correlation, and we conclude the experts may focus on different aspects of the patches that may be common to all classes (background, basic shapes, etc.).

To further investigate the logic behind the first layers of experts, Figure 30 shows the correlation between selected experts and the patch id or location. The model partitions each image in the same number of patches –say if the patch size is 14x14, and images are 224x224x3, then there are 256 patches (sometimes we add an additional learnable token). We add a positional embedding to each patch that helps the model track the relative ordering. In this case, we see that for the first few MoE layers, the routers tend to distribute patches to experts according to their patch id. One simple explanation could be that patches in similar positions usually share visual characteristics that one expert learns to handle –say, image corners, backgrounds, or the center of the images with objects.

### E.3 Routing weights distribution

Most of the key model hyper-parameters, like the number of experts that process each patch $k$ or the expert buffer capacity ratio $C$ that controls the amount of patch dropping, can be adjusted layer-wise. For example, if we do not see expert specialization in lower layers, we could simply set $k = 1$ there to avoid wasting compute. It may however be useful to increase $k$ in the last few layers to allow for composability of concepts, like when we try to identify several objects. Figure 31 shows the TOP-1 and TOP-2 weight distribution for an sparse model with $k = 2$. Two main conclusions can be drawn. First, in lower layers, both choices seem to have a similar magnitude –thus, both indeed contribute to the combined representation. Moreover, the weights are usually low in this layers –note $1/E \approx 0.03$ is the minimum weight the top selected expert can be assigned–, which one may interpret as the router being somewhat indifferent among experts. Second, the trend clearly changes when patches travel towards the end of the network. In particular, the TOP-1 and TOP-2 weight distributions strongly diverge, with the former approaching 1.0 and the latter approaching 0. This means the intrinsic $k$ at the top of the network is closer to 1 (than the actual $k = 2$). The composability that we mentioned before may not be indeed needed at the patch level, as patches are quite small for large networks (here, 14x14), and it may be difficult to identify several concepts. Nonetheless, some tail of the distributions shown in Figure 31 still uses both experts in the last layers.

We would like to remark that each *image* is subject to a large number of routing decisions through its patches. Concretely, Figure 32 shows how most images use –on aggregate by pooling over all their patches– most of the experts in *every* layer. This motivated our efforts to try to save compute by discarding, or not processing, patches that are not useful for the final classification. We cover this in detail in Section 4.

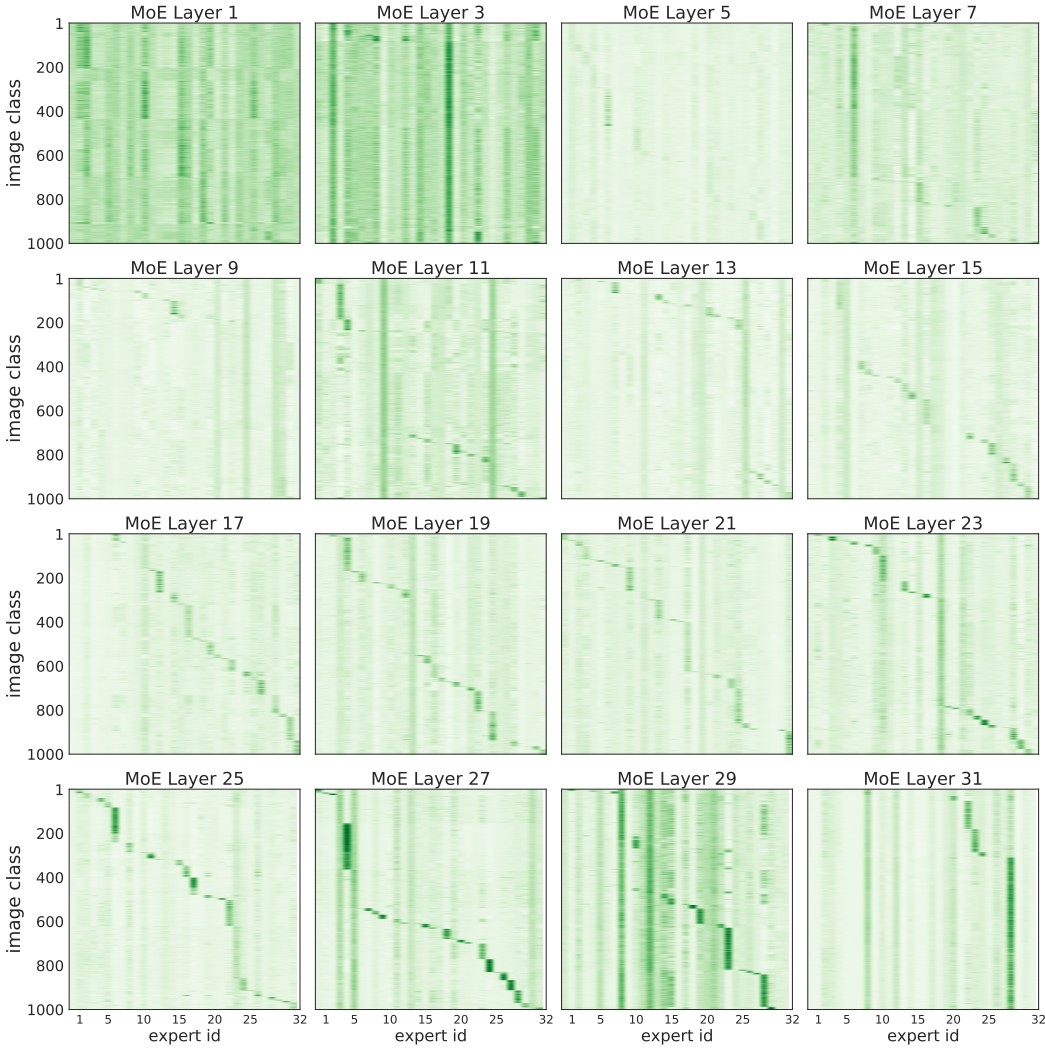

Figure 29: **Average weight for selected experts per class**. We show the 16 MoE layers of an every-2 V-MoE-H/14. The $x$-axis corresponds to the 32 experts in a layer. The $y$-axis are the 1000 ImageNet classes; orderings for both axes are different across plots. For each pair (expert $e$, class $i$) we show the average routing weight for the patches corresponding to all images with class $i$ for that particular expert $e$.

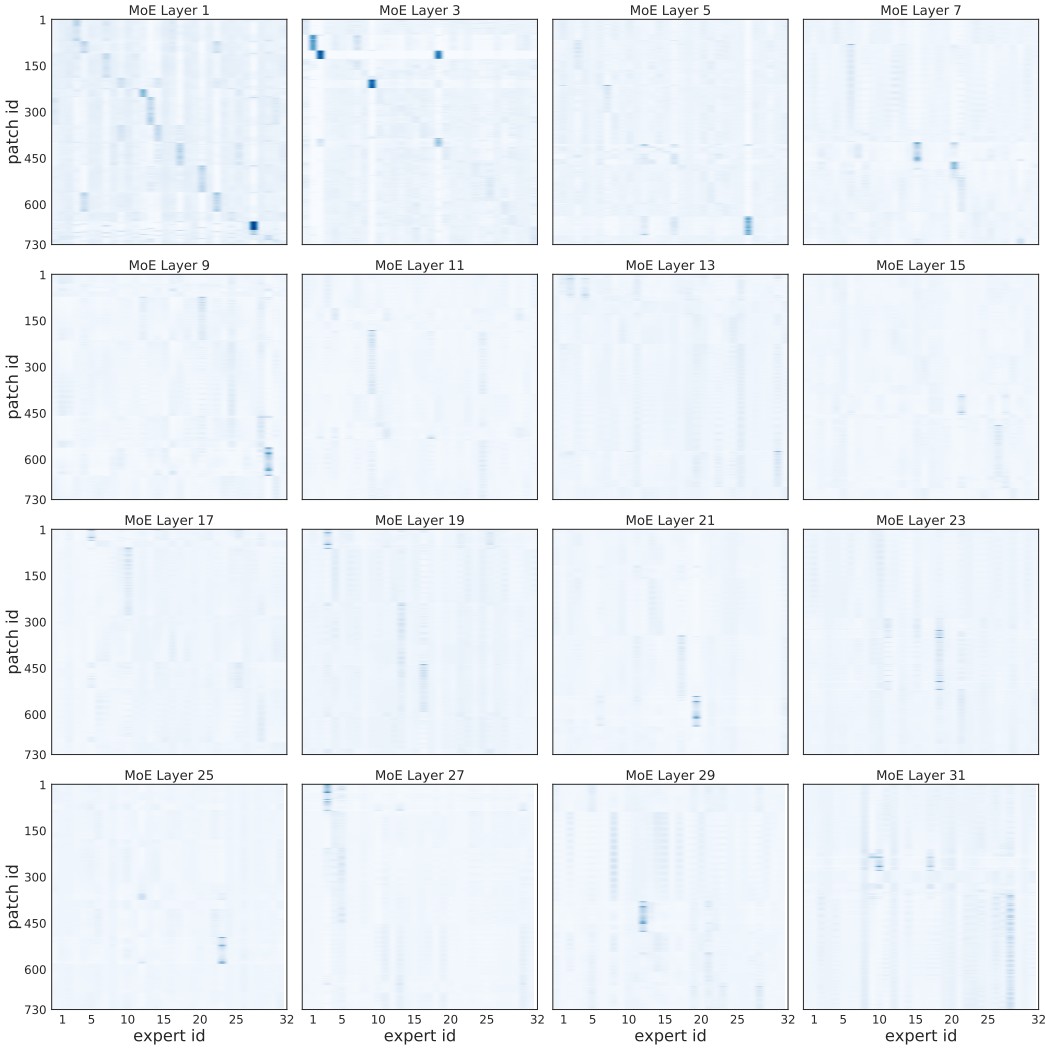

Figure 30: **Average weight for selected experts per patch position** on a every-2 V-MoE-H/14 fine-tuned model. The $x$-axis corresponds to the 32 experts in a layer. The $y$-axis are the 730 patches in ImageNet images with 14x14 patch size, at (384, 384, 3) resolution; orderings for the x-axis are different across plots. For each pair (expert $e$, patch-id $i$) we show the average routing weight for all the patches with patch-id $i$ that were assigned to that particular expert $e$.

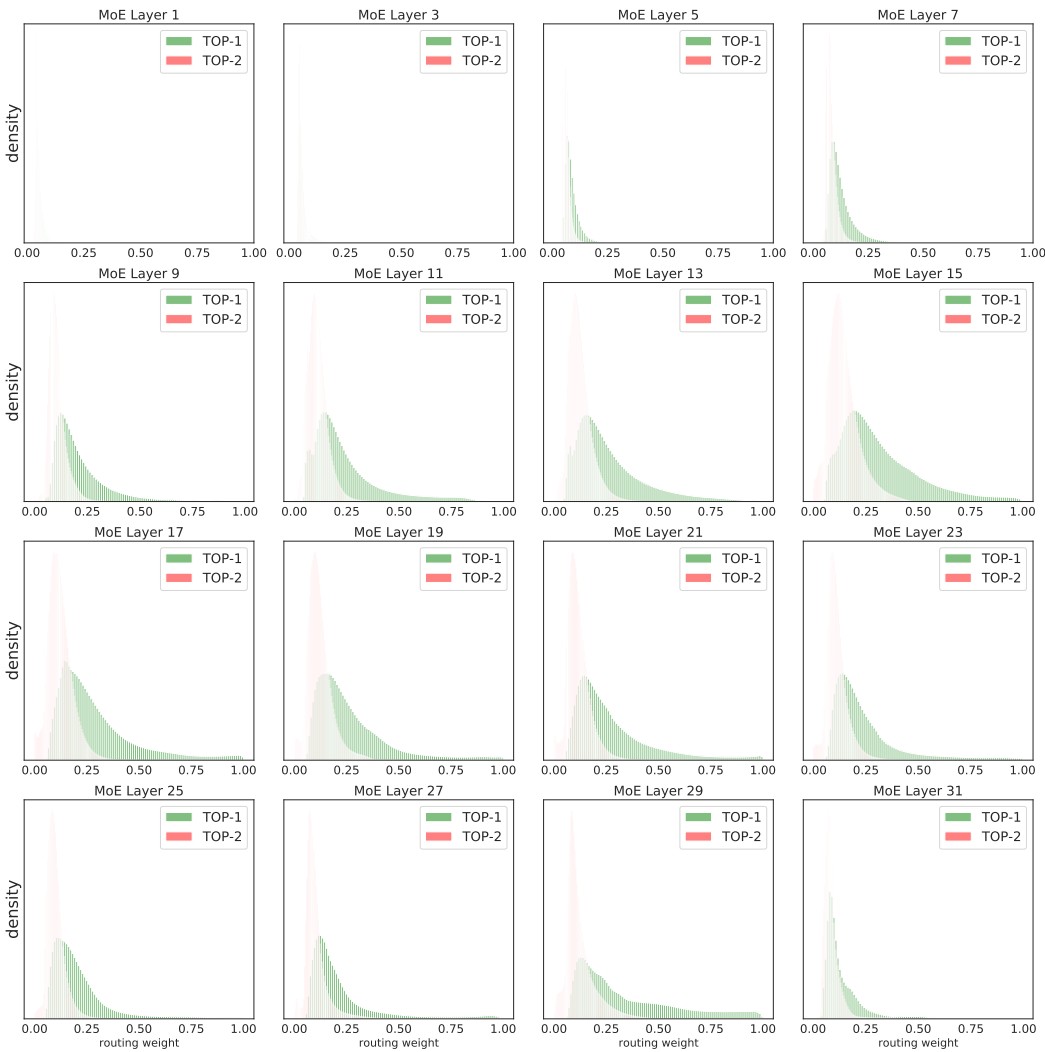

Figure 31: **Routing weight distribution for TOP-1 and TOP-2 selected experts.** We show the distribution over the TOP-1 (green) and TOP-2 (red) weights for a V-MoE-H/14 model fine-tuned on ImageNet. Note for any given patch these weights do not need to add to one —and in fact they will not—, as we apply the softmax before the TOP-$k$ selection.

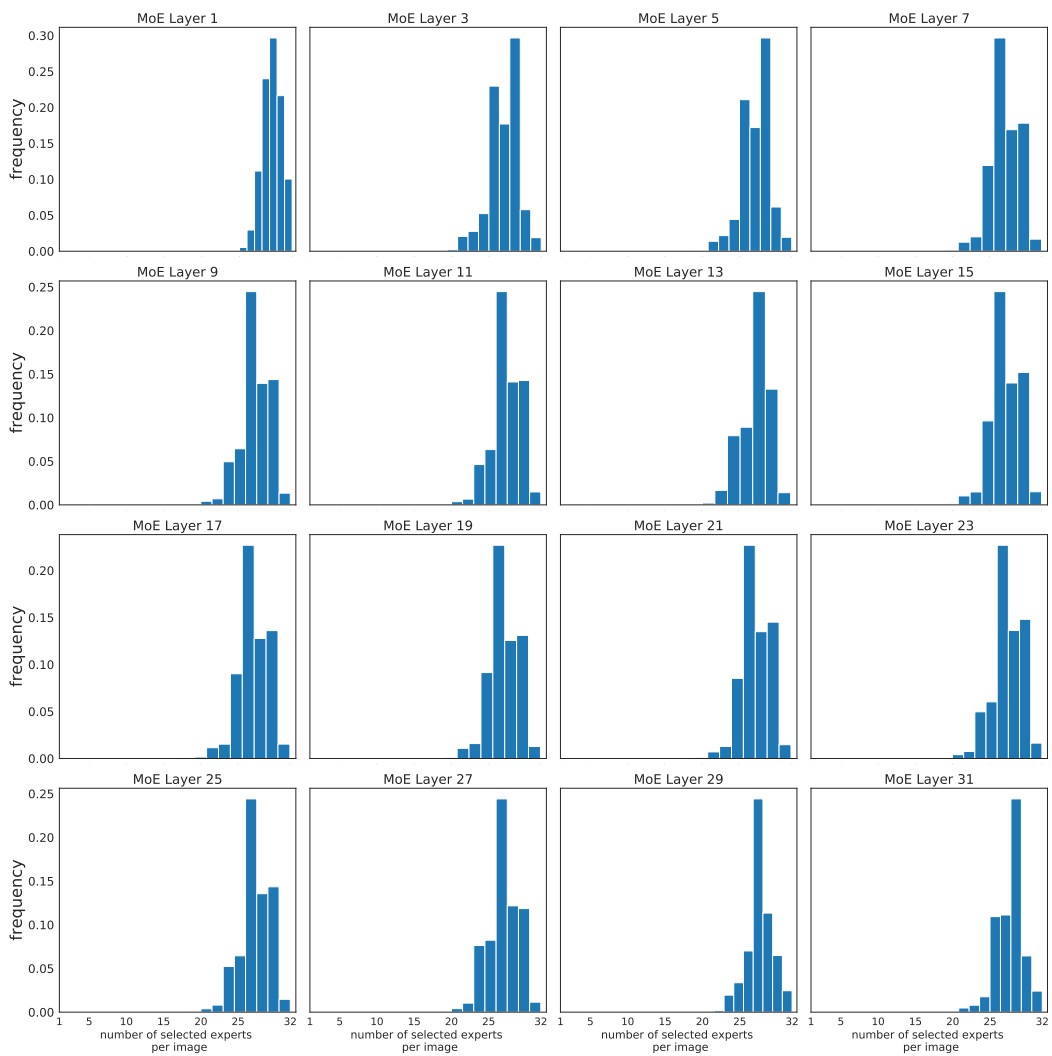

Figure 32: **Number of selected experts per image (after pooling selection from all patches).** We show the distribution of total number of used experts per layer per image for a V-MoE-H/14 model fine-tuned on ImageNet. In this case, every image has 730 patches. Even though most experts are selected at least once —that is what we plot here—, we expect some of the experts to be selected way more often by the patches of an image, and with a higher average weight.

### E.4 Changing $k$ at inference

We now explore a remarkable aspect of expert models: their flexibility. Somewhat surprisingly, we have observed sparse models to be fairly robust to mismatches between the training and inference configurations. In this section, we explore the effect of training with some original $k$ while applying the model at inference time with a different $k' \neq k$. This can be handy to control (decrease or increase) the amount of FLOPs per input in a particular production system.

Figure 33 is based on a V-MoE-S/32 model trained with $k = 1$. We evaluate the upstream and few-shot metrics at inference time for a range of new $k'$s. Note we do not perform any further training in any case, and the model parameters (including the router) are identical in all cases. The only difference is the number of experts we apply to each input, the amount of the network we activate. In red we show the original model's performance, and in blue the new ones. Finally, for each $k'$, in yellow, we show the performance of a V-MoE-S/32 model trained **originally** with $k = k'$ –which, as we expected, increases in $k'$. We see that increasing the value of $k$ from its original value ($k = 1$) at inference time by one or two units actually significantly improves performance, both upstream and few-shot. However, at some point, if the new $k'$ is too large, the performance starts suffering –probably as the model is not prepared for the new distribution of total output routing weights applied in the linear combination, and sub-optimal experts for a given input start contributing to its representation.

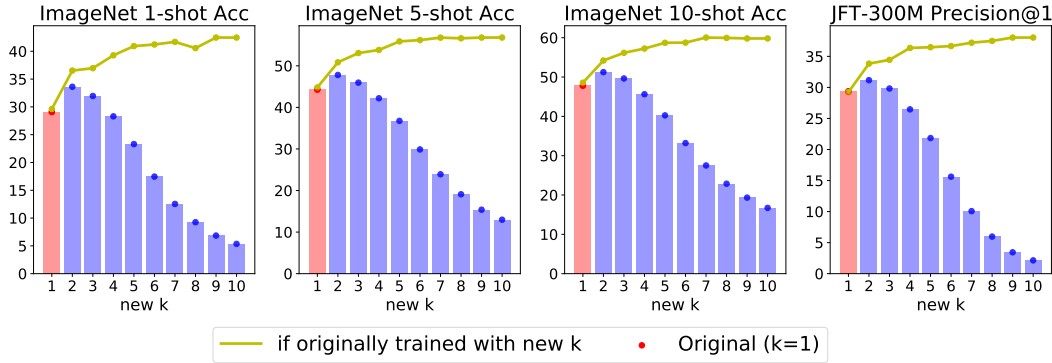

Figure 33: Original V-MoE-S/32 every-2 model was trained with $k = 1$.

Figure 34 shows the case where the original model is a V-MoE-S/32 with $k = 2$. The trends are somewhat similar. By applying $k' = 3$ or $k' = 4$ we obtain modest improvements, whereas by decreasing $k$ to $k' = 1$ we obtain a performance very similar to the performance of a model trained *directly* with $k = 1$, especially for few-shot. This is interesting, as we can devote more FLOPs for training by setting $k = 2$ upfront, while deferring the choice of inference $k$ without losing potential performance. We explored these ideas further in Section 4. Also, the drop in performance for large values of $k$ is less severe in this case, probably due to the fact that the trained model was used to combine several different experts (not the case for Figure 33).

Finally, in Figure 35 we present the case where the upstream model was trained with $k = 5$. This is an expensive model to train, and we see we can change the inference value of $k$ from $k' = 3$ to $k' = 7$ with results that are similar to their optimal value, if we had trained with those values in the first place. At this point the model is stable enough to deal with large values of $k$, but it suffers way more when we set $k' = 1$, as the model is not used to picking a single expert and –we suspect– the TOP-1 expert may not carry so much importance or weight for this model where five experts were selected per input while training. Of course, it may not just be a matter of routing weight distribution. The expert themselves may be quite different when training with $k = 1$ –say, more self-contained– and with $k = 5$ –perhaps more team-players.

### E.5 Changing $k$ during fine-tuning

We also consider the effect of adjusting the number of selected experts during fine-tuning and inference. We consider the aforementioned V-MoE-S/32 models, with 32 experts, pre-trained with

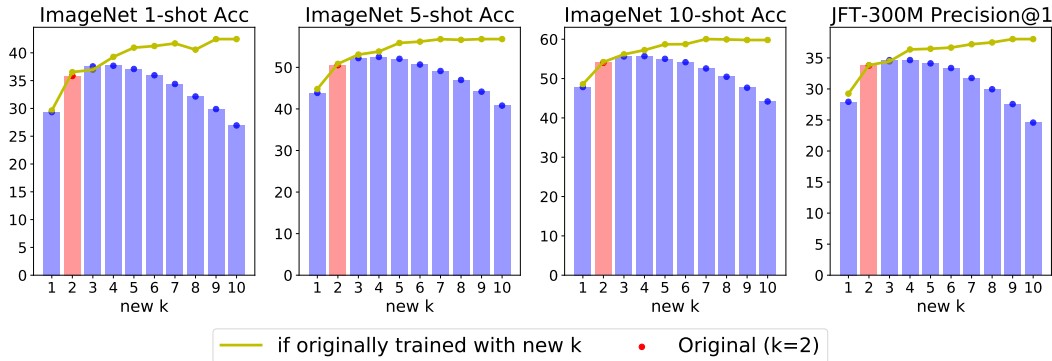

Figure 34: Original V-MoE-S/32 every-2 model was trained with $k = 2$.

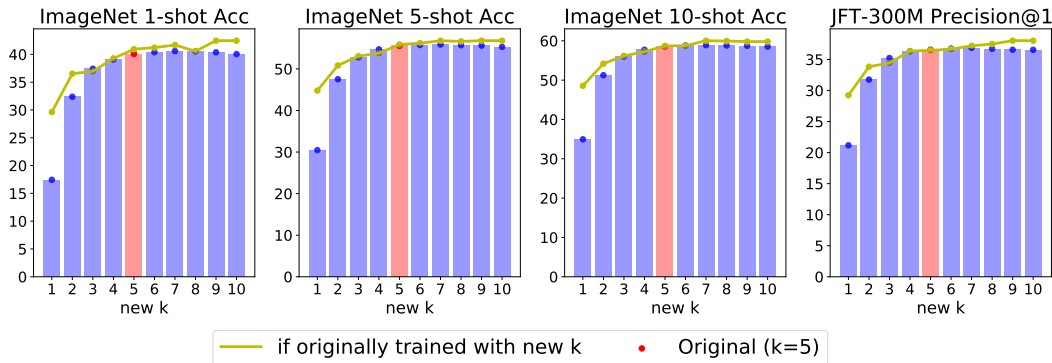

Figure 35: Original V-MoE-S/32 every-2 model was trained with $k = 5$.

$k = \{1, ..., 9\}$ experts. These models are then fine-tuned with varied $k$. We show the result of this in Figure 36. As one may expect given our previous results, generally increasing $k$ improves performance. Regardless of upstream $k$, generally accuracy improves from increasing $k$ during fine-tuning. Similarly, increasing $k$ during pre-training improves performance downstream.

Conversely, when $k = 1$ downstream, all models fail to improve from pre-training with higher upstream $k$. Models pre-trained with $k > 1$ seemingly learn to *combine* expert outputs, in that they do not generalize as well to selecting a single expert downstream, and lose the benefits of pre-training with larger $k$.

## E.6 Pre-training with less data

We have shown that the standard recipe of pre-training with large datasets allows use of powerful sparse models on downstream vision tasks where less data is available. The question naturally arises: do these models require large amounts of data upstream? We present here some initial explorations in this direction.

**Training on JFT300M with less data.** We first train a V-MoE-L/32 on subsets of JFT300M. This was also done for dense models in [20], and in Figure 37 we compare directly to their results. V-MoE seems initially fairly robust to reduced data, but after reducing to 9M pre-training samples (3% of the dataset), it becomes slightly preferable to instead train a dense model.

**Training on ImageNet21k.** ImageNet21k [16] is a large public dataset with approximately 14M images and 21k classes. Previous works [20, 36] have successfully pre-trained on it to achieve strong results in downstream tasks. In particular, dense ViT models trained on ImageNet21k perform reasonably well. With the exception of ViT-S, where V-MoE immediately outperforms the dense counterpart, applying sparse scaling generally harmed performance. We observed overfitting, both

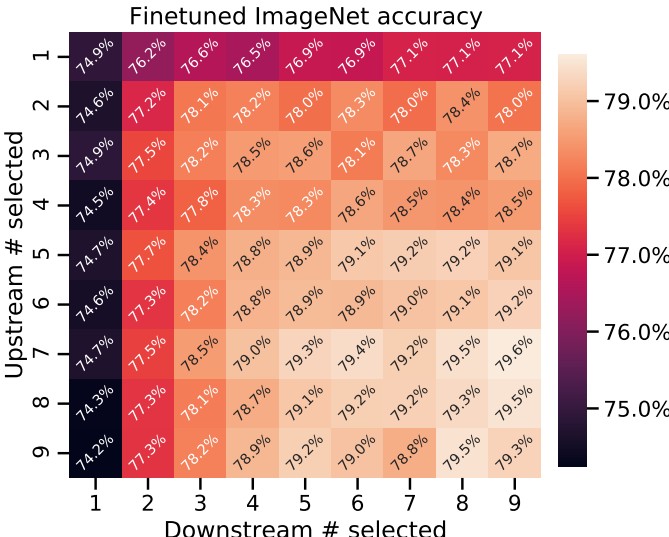

Figure 36: Varying $k$ (number of selected experts) at fine-tuning/inference times for V-MoE-S/32 models pre-trained with different values of $k$.

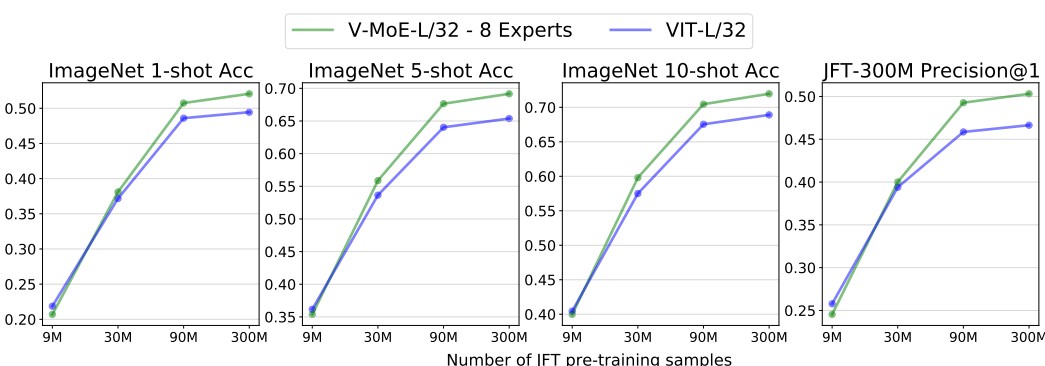

Figure 37: **The effect of varying the amount of pre-training data.** We compare the performance of V-MoE-L/32 and VIT-L/32 for increasing data sizes. In particular, we take subsets of JFT-300M with 9M, 30M, 90M, and 300M datapoints —note the full dataset contains around 305M datapoints. Given that we train with smaller datasizes, we decided to use 8 experts rather than 32 (every-2). At the lowest data size (9M, around 3% of the original), the MoE model is not able to leverage its extra-capacity. For the remaining ones, starting at 30M (around 10% of the original dataset), it does.

in the sense of reducing validation accuracy on the pre-training dataset, but also in reduced transfer performance as training continued. As an initial attempt at tackling this, we used RandAugment [14] with $N = 2$ transformations of magnitude $M = 10$. This is shown in Figure 38. Interestingly, RandAug typically helps expert models while harming dense models. With this applied, for each architecture, there is an expert model which outperforms the dense baseline.

This is far from a complete exploration; it indicates that these models can work with smaller data sources, and the key to their efficacy likely lies in more careful considerations of data augmentation and regularisation. We expect recent bodies of work exploring this for dense transformers [32, 60] to be useful here, and that works in data efficient vision transformers [59, 65] to also further unlock the potential of V-MoE with less pre-training data.

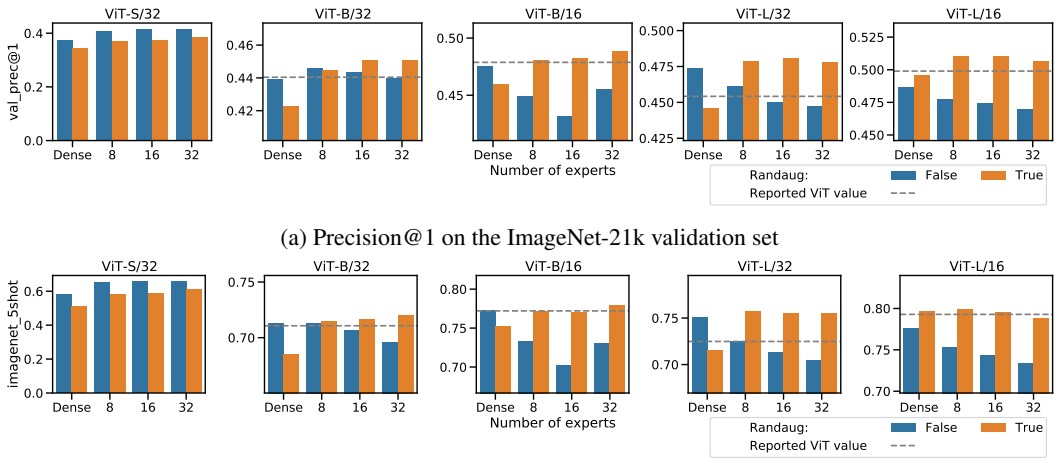

(a) Precision@1 on the ImageNet-21k validation set

(b) 5-shot linear ImageNet performance

Figure 38: **Performance of ImageNet-21k pre-trained models.**

## E.7    Effect of batch size

Tokens are distributed among experts, which are stored and run in parallel on separate devices. This naturally begs the question: Does this work at lower batch size? What happens if one example is processed and it wants to send the whole example to a single expert?

In order to explore this, we re-evaluate the 32 expert vMoE Every-2 models at different batch sizes.

First we show the inference latencies in Figure 39.

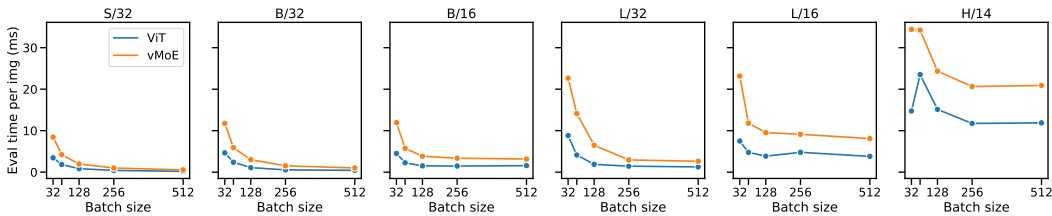

Figure 39: The effect of batch size on inference speed of V-MoE and ViT models. Numerical annotations above marked points indicate the batch size used to generate that point.

As expected, and observed there, all models suffer various inefficiencies and unamortized overheads at decreasing batch size. As vMoE models have much more cross-device communication, the fixed latency associated with such communications will be more significant at lower batch sizes and it is indeed worth noting that this may make it scale down worse than the fully dense ViT, but it does not seem to be a significant issue here.

A separate but related concern for V-MoE models is that the batch size interacts with balancing of the load and distribution of tokens - at lower batch sizes, it is more likely that all the images in a batch will want to use the same experts, leading to oversubscribed experts and dropping of tokens. Therefore we want to show not just the interaction between batch size and latency, but also its effect on performance. We have plotted this in Figure 40.

As expected batch size doesn't impact dense ViT models but does effect performance for V-MoE. However, the sparse models still perform well, with a higher performance for a given latency.

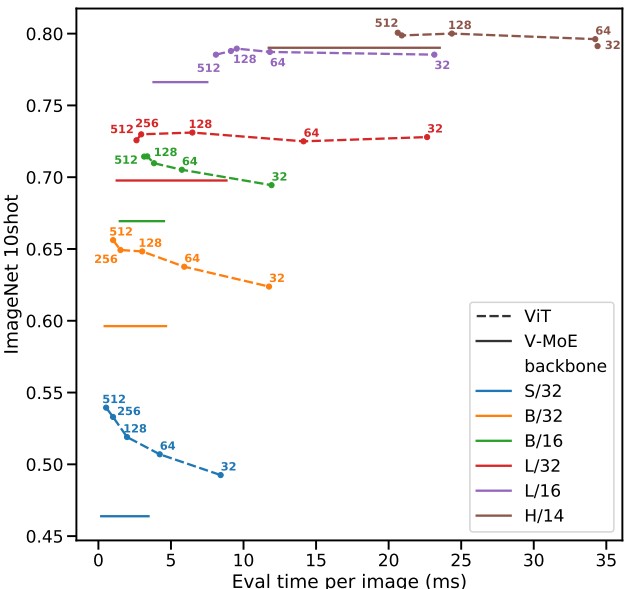

Figure 40: The effect of batch size on ImageNet-10shot performance of V-MoE Every-2 models.