# OpenReview forum: "Scaling Vision with Sparse Mixture of Experts"
_NeurIPS.cc/2021/Conference — NeurIPS 2021 Poster_

### Official Review · Reviewer_fg9p · 2021-07-17

**Rating:** 6
**Confidence:** 4

**Summary:**

This paper proposes an approach to scale vision transformers by using sparse mixture of experts. The idea is simple and straightforward---using a subset of experts defined by a topk operation. The authors conducted sufficient experiments on ImageNet and the experiments demontraste the approach can achieve a nice tradeoff between accuracy and comptutational cost.

**Limitations And Societal Impact:**

Yes

**Main Review:**

Strengthes:
1. The idea is simple and straightforward.
2. Sufficient experiments are analysis are provided.

Weaknesses:
1. The presentation of this paper could be further improved. For instance, Figure 1 is not intuitive and straightforward. In L81, the authors defined the rounting function $g(x) = Top_k(softmax(Wx + \epsilon))$. What does $x$ mean here? a patch representation? In 208, the routing function is defined to operate on $X$. So the notations throughout the paper are rather confusing.

2. How do you optimize top_k since this would be a non-differentiable operation?

3. In figure 1, what does device mean? different gpus? different computation nodes?

4. The authors mentioned that "The difference between previous formulations [38] is that we apply TOPk after the softmax over experts weights, instead of before". If you select top k experts and only use them for computation, you can save computation. I am not sure how could you save computation if you use all experts first, and then select topk experts?

5. It would be better to explictly discuss wow the MoE layer is different from a standard MLP, e.g., describing their weight dimensions.


**Time Spent Reviewing:**

4

---

> ### Author Response · Authors · 2021-08-09
> **Initial author response.**
>
> First, we would like to sincerely thank the reviewer for their time and constructive feedback.
>
> Addressing the comments in turn:
> 1. Regarding notation, we apologize for the confusion. We use vector and matrix notation for a vector **x** and a matrix **X**. In L84-85, **x** is defined to be the representation of a single image token at some layer of the network, and also in L63. **X** is simply a batch of **x** and is defined in L208 (“the routing function g is applied row-wise to a batch of inputs **X**”). If there are suggestions on clearer notation, we are happy to make those changes. In principle, the router can either operate on single patches or on batches of patches. However, both at training and inference, we usually operate on a set of patches (coming from one or multiple images).
>
> 2. Good point. Similar to previous works, we do not differentiate through the top-k operation. However, the router gets gradients through other paths:
>     * the combination of MoE outputs, which is indeed weighted by the router weights.
>     * the auxiliary loss, that uses all the routing weights.
>
>     The first one can be considered a proxy for directly optimising the routing behaviour - if the model wants to keep an expert’s output with a high weight in the combination function, then it is sensible to route to that expert. The second favours some specific types of routing (balanced ones) without explicitly accounting for task performance.
>
> 3. In practice, a device means different TPU cores. More abstractly, a device can mean any different computation node. This could mean a different GPU; for example, recent works have successfully implemented large sparse models on 480 Nvidia GPUs [1].
>
> 4. We do not use all the experts. The quoted text refers to the expert *weights*, which are outputs of the router, before applying any expert. For example, given 4 experts to choose from and k = 2, suppose the router outputs [1, 2, 0, -1]. We pick the top two experts, which are the first two. If we apply top-k before the softmax, as in previous works, then the routing weights for those two experts become [0.27, 0.73] i.e. they add up to 1. In our formulation, we softmax before top-k, leading to [0.24, 0.64, 0.09, 0.03], so the selected routing weights are [0.24, 0.64]. After computing the weights this way, then we apply the selected experts (and average their outcomes according to the weights). We will reword this to explain it better in the paper.
>
> 5. A dense model implements MLP(x) = Dense(gelu(dense(x))). An MoE layer implements a mixture of the exact same MLP (though each with different weights). For example, when there are 32 experts, we have 32 such MLPs in parallel (each with different weights). The router predicts coefficients which are used to combine the outputs of multiple such MLPs, e.g. the top 2 for a given example. For N experts, the MoE layer thus consists of N MLPs, and a router which is a single dense layer projecting to size N. We will improve the explanation in the paper.
>
> [1] Exploring Sparse Expert Models and Beyond, A. Yang et al, https://arxiv.org/abs/2105.15082

---

> > ### Comment · Reviewer_fg9p · 2021-08-30
> > **Re:Initial author response**
> >
> > I appreciate the response from the authors and my concerns are addressed. I'd like to update my score and recommend acceptance.

---

### Official Review · Reviewer_mFJK · 2021-07-17

**Rating:** 7
**Confidence:** 3

**Summary:**

The paper extends the sparsely-gated mixture of experts networks for NLP to the applications in the computer vision. A batch prioritized routing technique is proposed to discard the least useful patches to reduce the buffer size and computational cost. From the experimental results, the proposed method has shown improvements in large vision models and representation learning.

**Ethical Concerns:**

No.

**Limitations And Societal Impact:**

Yes.

**Main Review:**

1. Originality:  The paper is an extension of the works of "sparsely-gated mixture-of-experts layer", "switch transformers”， and "scaling giant models". A batch prioritized routing technique is proposed to improve the performance. The work is more technically sound than the novelty.

2. Quality: The work is technically sound. The experimental results are sufficient and shown improvements in large vision models and representation learning.

3. Clarity: The paper is well written and organized. The paper is easy to follow.

4. Significance: The result has referential significance. It is not a new idea, but the work is technically sound.

**Time Spent Reviewing:**

12

---

> ### Author Response · Authors · 2021-08-09
> **Initial author response.**
>
> First, we would like to sincerely thank the reviewer for their time and constructive feedback.
>
> We are glad to see the quality of writing, significance of results and thoroughness of experiments were appreciated. Though there are no specific misunderstandings to clarify or points to discuss, we would like to address the point surrounding novelty and significance. As noted in the review, the BPR technique developed here is novel, which can directly be applied back to sparse NLP models for computational efficiency and robustness to dropping of tokens. As well as technical contributions, we believe the extensive analysis and ablations provide insight thus far not explored in the domain of NLP.
>
> More philosophically and higher level, it is arguable that many major breakthroughs in deep learning consist of reapplying ‘not new’ ideas (e.g. an existing model) to a new domain: for example, the application of LSTMs to Machine Translation, CNNs to TTS, or indeed Transformers to Vision. In all cases, including V-MoE, getting the model to work very well in a completely novel domain is very difficult (e.g. MoE transformers can be unstable and hard to finetune [1]). However, it is very valuable: once the method has been made to work, as we have done here, the community in the target domain (here: computer vision) can more easily build on the technique.
>
> [1] Switch Transformers: Scaling to Trillion Parameter Models with Simple and Efficient Sparsity, Fedus et al. https://arxiv.org/abs/2101.03961

---

### Official Review · Reviewer_Zhow · 2021-07-18

**Rating:** 9
**Confidence:** 5

**Summary:**

This paper trained the current largest vision model through MoE techniques. It achieved strong top-1 accuracy on ImageNet-1K image recognition, at 90.25%. It also propose batch prioritized routing. When working together with a C << 1, it can further largely reduce the computation.



**Ethics Review Area:**

["I don’t know"]

**Limitations And Societal Impact:**

-- The sparse MoE algorithm is efficient in inference only when run in batch mode. When there is only one sample or a few samples, it will be unpractical. This could prevent its further prevalence.

-- In fig.3, it seems meet overfitting problem when the model is as large as H/14. Can the authors explain more on it please?

-- In Line 86-88, it gives a impression that the routing after softmax is novel. However, Gshard and Switch Transformer have used this technique. It is better to refer to Gshard and Switch Transformer when introducing this.

- The expert number is fixed as 32. What is the performance at more experts, e.g. 64, 128, 256, or less experts, e.g. 16, 8 and etc.

**Main Review:**

Originality

-- Sparse MoE has proved working well in NLP, and so it is not strongly original. Nevertheless, I see the achievement of successfully adapting this model to the field of computer vision as adequate originality. To the technical part itself, batch prioritized routing is novel, and seems generalizable to other problems. It will be interesting to see its performance in NLP tasks.

Quality

-- The designs seem reasonable and make sense to me. The experiments are also rich. It is a submission of high-quality.

Clarity

-- very clear and well written.

Significance

-- Strong as I can tell. The difficulty of deployment may block its higher interest to the community. Nevertheless, this is a problem of the sparse MoE technique, but not refers to the specific vision applications.

**Time Spent Reviewing:**

6 hours

---

> ### Author Response · Authors · 2021-08-09
> **Initial author response.**
>
> First, we would like to sincerely thank the reviewer for their time and constructive feedback.
>
> Moreover, we are extremely thankful to the reviewer for highlighting the originality, quality, clarity and significance of our work. Here are some comments regarding the limitations that have been brought up:
>
> 1. *“MoE algorithm is efficient in inference only when run in batch mode”.*
>
>     Note that the routing is done on individual patches. For the image and patch sizes typically used by practitioners, this usually gives sequences of over 100 patches. Therefore even with just one image, there’s typically enough load to keep the 32 experts busy. Considering also that (i) individual images tend to use a diverse range of experts (see Figure 32) and (ii) the model is robust to some patch dropping, we expect that many of the efficiency gains can still be achieved at low batch sizes.
>
> 2. *“Overfitting problem when the model is as large as H/14”.*
>
>     We do not observe “overfitting” in the classical sense (ImageNet training loss going down while validation loss going up). As for diminishing returns in a transfer learning sense, we acknowledge but do not have a clear understanding of the observed behaviour. Nevertheless, we tried a few regularization techniques during fine-tuning such as applying dropout (throughout the model and only on experts) and random augmentation – these can yield a bit of a performance boost. There are also MoE specific downstream aspects (whether to include the auxiliary loss or not, increase or decrease the router noise etc). Although it is possible to squeeze more transfer performance by tuning such things, we believe more significant progress will be made with adaptation methods more tailored to sparse models, e.g. ‘pruning’ experts that are not useful for a new task. This is still ongoing work.
>
> 3. *“In Line 86-88, it gives the impression that the routing after softmax is novel”.*
>
>     We’ll rephrase the paragraph including references to GShard and Switch.

---

> > ### Comment · Reviewer_Zhow · 2021-08-16
> > **follow-up question**
> >
> > ------ "Therefore even with just one image, there’s typically enough load to keep the 32 experts busy"
> >
> > Can you provide some the real latency with increasing image number per batch (e.g. 1, 2, 4, 8, 16)？
> >
> > ------“The expert number is fixed as 32. What is the performance at more experts, e.g. 64, 128, 256, or less experts, e.g. 16, 8 and etc.”
> >
> > I did not find answers to address this question.

---

> > > ### Author Response · Authors · 2021-08-20
> > > **Follow-up answers**
> > >
> > > *1) Batch size & latency*
> > > There are a lot of nuances to be aware of here so excuse us for the long response!
> > >
> > > First, a quick note: In theory expert models can be stored in one device, but in our infrastructure each expert lives in a different device. Therefore there is no point running a batch size less than # of experts (and our infra doesn’t support it).
> > >
> > > We re-evaluated the 32 expert vMoE models (experts every 2 layers) at different batch sizes.
> > > First we show the inference latencies: [Figure](https://i.imgur.com/NVYPRrt.png).
> > >
> > > As expected, and observed there, all models suffer various inefficiencies and unamortized overheads at decreasing batch size. As vMoE models have much more cross-device communication, the fixed latency associated with such communications will be more significant at lower batch sizes and it is indeed worth noting that this may make it scale down worse, but it does not seem to be a significant issue here.
> > >
> > > A separate but related concern here for expert models is that the batch size interacts with balancing of the load and distribution of tokens - at lower batch sizes, it is more likely that all the images in a batch will want to use the same experts, leading to oversubscribed experts and dropping of tokens*.
> > >
> > > Therefore we want to show not just the interaction between batch size and latency, but also its effect on performance. We have plotted this [here](https://i.imgur.com/7D5n5N3.png).
> > > As expected batch size does impact performance for vMoE models but not ViT models. However, the sparse models still perform well, with a higher performance for a given latency.
> > >
> > > We will add these figures to the paper, as we believe this is an important point for future works on sparse conditional computation to consider.
> > >
> > > *2) Changing number of experts*
> > > Unfortunately retraining models is very expensive, so we cannot answer this question with concrete numbers and can only provide anecdotal observations. It varied depending on architecture but we generally found there was no immediate advantage from using more than 32 experts. We are actively working on understanding why this is the case, and how to make use of more experts.

---

### Official Review · Reviewer_pW2k · 2021-07-19

**Rating:** 7
**Confidence:** 5

**Summary:**

This paper presents a scaling strategy for vision transformer using a widely-used approach, Mixture of Experts (MoEs) on Feed-Forward Networks. The simple scaling strategy could achieve a high performance of 90.2% top-1 accuracy with a 15B parameter vision transformer. Interestingly, V-MoE merges FFN-MoE with a spatial dynamic computation algorithm, to decide whether each patch would be computed in each layer and reduce the computation in the inference stage with few performance drop.

**Ethical Concerns:**

My ethical concerns mainly lie on the data side. The JFT-3M and JFT-3B datasets are not publicly available, which should be checked that whether they contain the images in the ImageNet validation set, and whether exists ethical issues inside the datasets.

**Ethics Review Area:**

["Privacy and Security (e.g., consent)"]

**Limitations And Societal Impact:**

Yes.

**Main Review:**

For originality, this paper extends the existing techniques (MoEs) in NLP to Vision Transformer, and also studies how to scale up vision transformer in an empirical manner. The combination of MoE and spatial dynamic computation is interesting and smart. The experimental study of this paper is valuable to the community.

The quality of this paper is good. From an experimental perspective, this paper contains sufficient experimental results, verifies important claims, and achieves state-of-the-art performance on ImageNet even with a significantly large number of parameters.

Concerns:
1. Some study has found many of the labels of ImageNet validation set are incorrect (“Pervasive Label Errors in Test Sets Destabilize Machine Learning Benchmarks”). I wonder whether the 90.2% accuracy is truly correct or somewhat overfit the validation set. Or it would be better to see the real upper bound of the accuracy on the ImageNet validation set or how the models perform on ImageNet-V2 validation set.
2. Data leaking of the Imagenet-1K validation data is also a potential issue since the de-duplication method is not so reliable when the data number is extremely large (e.g. 3B).

**Needs Ethics Review:**

Yes

**Time Spent Reviewing:**

5

---

> ### Author Response · Authors · 2021-08-09
> **Initial author response.**
>
> First, we would like to sincerely thank the reviewer for their time and constructive feedback.
>
> We address the comments below.
> 1. We understand and share concerns about benchmark reliability. Previous works show that higher ImageNet performances correlate very strongly with better transfer and general relevance of features [1]. The strong few-shot and transfer learning performance reported here lends credibility to claims of model competence – the setup has not overfitted to the ImageNet validation set as it performs well when adapted to a variety of tasks (ImageNet, CIFAR, Pets etc) with different adaptation methods (linear few-shot or fine-tuning).
>
> 2. For few-shot and full finetuning, we use high-recall deduplication of the pre-training data (JFT300M, JFT3B and ImageNet21k) as done in previous works [2]. The deduplication was found to have minimal impact, indicating that data leakage does not contribute significantly to results. This agreed with our own explorations of data deduplication (see Table 9).
>
> [1] Do Better ImageNet Models Transfer Better, S. Kornblith et al, CVPR 2019.
> https://openaccess.thecvf.com/content_CVPR_2019/html/Kornblith_Do_Better_ImageNet_Models_Transfer_Better_CVPR_2019_paper.html
>
> [2] Big Transfer (BiT): General Visual Representation Learning, Kolesnikov et al, ECCV 2020
> https://link.springer.com/chapter/10.1007/978-3-030-58558-7_29

---

### Review · Ethics_Reviewer_17Tc · 2021-08-09

**Recommendation:**

I do not think this paper should be rejected for using JFT-300M or JFT-3B since these are reasonably established datasets and the question of their use should not be assigned to a single set of authors. However, I would encourage the authors to provide (or encourage the owners of the dataset to provide) more information regarding how the data was generated, what type of consent was acquired and how the dataset was reviewed to ensure problematic images were removed from the dataset. I also encourage them to read and address the questions raised in this paper: https://openreview.net/pdf?id=s-e2zaAlG3I.

**Ethics Review:**

The ethical concern is that the data used in the paper is JFT-300M and JFT-3B. These are internal Google datasets that are not publicly available. These datasets are reasonably well established and there at least 40 other published papers that use JFT-300M. While the lack of transparency regarding these datasets is concerning (see recommendations below), I don't think it warrants rejecting this particular paper for ethical concerns.

---

> ### Author Response · Authors · 2021-08-20
> **Acknowledgement + response**
>
> Many thanks for taking the time to review the ethics of our paper! It is seriously appreciated.
>
> Firstly, apologies for the omissions in our ethical statements. We were unsure when considering ethical aspects whether to only focus on areas we have ‘ownership’ over. Though it is important for researchers to consider the ethics of *any* components they interact with we were wary about effectively introducing templated ethical statements about ImageNet/JFT which would be repeated word-for-word with many other publications that operate in this domain/paradigm.
>
> We understand the concerns relating to privacy, consent & reproducibility and aim to discuss them with the owners of the JFT dataset/pipeline.
>
> In regards to reproducibility, we have presented some ImageNet21k models. They were admittedly a small part of this effort. We are actively working on applying [recent work](https://arxiv.org/abs/2106.10270) in order to improve the standard of these models with the hopes of open sourcing pre-trained vMoE models, not only for reproducibility’s sake but also to encourage experimentation with sparse models in the computer vision community.

---

### Review · Ethics_Reviewer_1fGj · 2021-08-11

**Recommendation:**

Considerations for ethical review and research could include how these models are leveraged for systems like facial recognition, in the workforce, potentially for remote proctoring and with other forms of machine learning algorithms. It would be interesting if the paper ventured to tackle one of these real-world implications and moved to introduce an analysis on how to address it.

**Ethical Issues:**

Yes

**Ethics Review:**

The authors employed sparse conditional computation to train large vision models. They present a Vision MoE (V-MoE), a sparse version of the Vision Transformer, that is scalable and competitive with the largest dense networks. The paper solves for time and performance and concludes on hope to potentially reduce environmental impact. It does not dive deep into these environmental implications.

---

> ### Author Response · Authors · 2021-08-20
> **Author response**
>
> We very much appreciate the time taken reviewing the ethical aspects of this effort.
>
> We have taken the comments on board but wanted to ask a quick question concerning *the potential [of scaling] to further exacerbate existing biases* - does the reviewer know of any works that demonstrate this? We know of the [stochastic parrots](https://dl.acm.org/doi/10.1145/3442188.3445922) effort, but many of the concerns raised there are specific to generative models and NLP. On the other hand, we know for example of [efforts showing that scaling models down can result in unfair bias](https://arxiv.org/abs/2010.03058) and similar works. If there is anything computer vision specific that can be recommended here, we'd love to hear it!

---

> > ### Comment · Ethics_Reviewer_1fGj · 2021-08-28
> > **Response to feedback on Paper9164 from Authors**
> >
> > Thank you for your response to my ethics review.
> >
> > I agree with the recommendations identified by Ethics Reviewer 17Tc and am moved to identify the same paper for review and response. Here is a related article on the subject: https://analyticsindiamag.com/image-datasets-bias-privacy-mit/.
> >
> > In addition to that, I further suggest that scaling up existing models that are trained with little transparency, make it difficult to promise ethical application. I acknowledge that solving the societal implications addressed under my ethical review are not the responsibility of the authors. However, I would argue that since these societal implications have already impacted the real world, that the authors would benefit from addressing at least one of them in their paper and proposing how they would work to minimize those societal impacts in their scaled up design.
> >
> > The ask is that the authors approach the research with the understanding that data is not neutral, technology is not neutral, and therefore its consequences cannot be neutral, especially when trained on data that cannot be discerned.

---

### Decision · Program_Chairs · 2021-09-28

**Decision:**

Accept (Poster)

**Comment:**

All the reviewers agree that this submission made a significant contribution to the community. The combination of MoE and spatial dynamic computation is interesting and smart. The experimental study of this paper is valuable to the community. Although combining MoE to transformer has been studied in the NLP community and thus the originality of this submission is kind of limited, the submission still makes valuable contributions as mentioned by the reviewers. AC has read the submission, reviews and discussion and agrees with the reviewers on their recommendation.

**Consistency Experiment:**

NeurIPS has a long history of experimentation. In 2014, NeurIPS ran an experiment in which 10% of submissions were reviewed by two independent committees to quantify the randomness in the review process. This year, we repeated a variant of this experiment to see how the quality of the review process has changed over time.  This paper was part of the experiment and was therefore assigned to two committees (consisting of reviewers, an Area Chair, and a Senior Area Chair) that reached independent decisions.  If both committees made the same recommendation, this recommendation was followed. If a single committee recommended acceptance, the paper was accepted (with the exception of a few cases in which the other committee identified what we considered a fatal flaw, e.g., an error in a key result).

Both committees reached the same decision: **Accept (Poster)**

The other committee assigned to the paper recommended **Accept (Poster)**.  You can find the other set of reviews, along with any follow up discussion with the authors here:
https://openreview.net/forum?id=FrIDgjDOH1u